# Mixing layer height as an indicator for urban air quality?

Alexander Geiß[1,a], Matthias Wiegner[1], Boris Bonn[2,b], Klaus Schäfer[3], Renate Forkel[3], Erika von Schneidemesser[2], Christoph Münkel[4], Ka Lok Chan[1], and Rainer Nothard[5]

[1]Ludwig-Maximilians-Universität (LMU Munich), Meteorological Institute, Theresienstraße 37, 80333 München, Germany
[2]Institute for Advanced Sustainability Studies (IASS), Berliner Straße 130, 14467 Potsdam, Germany
[3]Karlsruhe Institute of Technology (KIT) – Institute of Meteorology and Climate Research, Atmospheric Environmental Research (IMK-IFU), Kreuzeckbahnstraße 19, 82467 Garmisch-Partenkirchen, Germany
[4]Vaisala GmbH, Weather Instruments, Notkestraße 11, 22607 Hamburg, Germany
[5]Senate Department for Urban Development and the Environment, Brückenstraße 6, 10179 Berlin, Germany
[a]now at: ESA ESTEC, Keplerlaan 1, 2201AZ Noordwijk, The Netherlands
[b]now at: Albert-Ludwig-University, Institute for Forest Sciences, Georges-Köhler-Allee 53, 79110 Freiburg, Germany

*Correspondence to:* Matthias Wiegner
(m.wiegner@lmu.de)

**Abstract.** The mixing layer height (MLH) is a measure for the vertical turbulent exchange within the boundary layer, which is one of the controlling factors for the dilution of pollutants emitted near the ground. Based on continuous MLH measurements with a Vaisala CL51 ceilometer and measurements from an air quality network, the relationship between MLH and near surface pollutant concentrations has been investigated. In this context the uncertainty of the MLH retrievals and the representativeness of ground-based in-situ measurements are crucial. We have investigated this topic by using data from the BAERLIN2014 campaign in Berlin, Germany, conducted from June to August 2014. To derive the MLH three versions of the proprietary software BL-VIEW and a novel approach COBOLT were compared. It was found that the overall agreement is reasonable if mean diurnal cycles are considered. The main advantage of COBOLT is the continuous detection of the MLH with a temporal resolution of 10 minutes and a lower number of cases when the residual layer is misinterpreted as mixing layer. We have calculated correlations between MLH as derived from the different retrievals and concentrations of pollutants ($PM_{10}$, $O_3$ and $NO_x$) for different locations in the metropolitan area of Berlin. It was found that the correlations with $PM_{10}$ are quite different for different sites without showing a clear pattern, whereas the correlation with $NO_x$ seems to depend on the vicinity of emission sources in main roads. In case of ozone as a secondary pollutant a clear correlation was found. We conclude that the effects of the heterogeneity of the emission sources, chemical processing and mixing during transport exceed

the differences due to different MLH retrievals. Moreover, it seems to be unrealistic to find correlations between MLH and near surface pollutant concentrations representative for a city like Berlin (flat terrain), in particular when traffic emissions are dominant. Nevertheless it is worthwhile to use advanced MLH retrievals for ceilometer data, e.g. as input to dispersion models and for the validation of chemical transport models.

## 1  Introduction

Air pollution is one of the major environmental issues in metropolitan areas because of its adverse effects on human health (e.g. Chen and Kann, 2008; Rückerl et al., 2011; Lelieveld et al., 2015). Strong emissions, e.g. from traffic, industry or heating, can drastically decrease air quality in particular when the emitted pollutants are captured below an inversion, and when meteorological conditions prevent an exchange of polluted and clean air. Without effective vertical mixing and advection pollutants can accumulate in the lowermost atmospheric layers and concentration thresholds as defined e.g. by the European Union Air Quality Standards (Directive 2008/50/EC) may be exceeded. For this reason several trace gases and particle mass concentrations (diameter below 10 $\mu$m, $PM_{10}$) are continuously monitored by air pollution monitoring networks near the surface implemented by federal or state administrations. In case of an exceedance of legally binding thresholds measures to reduce pollution are mandatory. This could e.g. include restrictions for motorized individual traffic.

Surface concentrations of gaseous pollutants as nitrogen oxides ($NO_x$), ozone ($O_3$), sulfur dioxide ($SO_2$) or carbon monoxide (CO) as well as particulate matter are routinely measured by in situ monitoring stations. Gaps of in-situ measurement networks can be filled by data from remote sensing techniques (e.g. Gupta et al., 2006; Martin, 2008) or numerical models. To better understand – or supplement – direct observations, air quality may be linked to integral parameters such as the aerosol optical depth (e.g. Koelemeijer et al., 2006; Schäfer et al., 2008; Li et al., 2016) or to meteorological parameters such as the height of the mixing layer (henceforward referred to as MLH or $H_{ml}$). The MLH can be considered as a measure for the vertical mixing within the atmospheric boundary layer and determines the dilution of pollutants which are emitted near the ground. Therefore, the MLH is frequently examined in evaluation studies of regional chemistry transport models (LeMone et al., 2013; Scarino et al., 2014; Brunner et al., 2015; Kuik et al., 2016) or serves as an input parameter for chemical box models (Knote et al., 2015). Due to the close relationship between turbulent vertical exchange and near surface air quality, several attempts have been made to establish correlations between MLH and near-surface pollutant concentrations (examples will be given in section 2). The underlying assumption is that high concentrations close to the surface may coincide with shallow mixing layers and vice versa. This assumption, which is used although vertical mixing is certainly not the only controlling process (e.g. Elminir, 2005; Tandon et al.,

2010; Svensson et al., 2011), will be examined in this paper. Our study is based on two months of data in summer from the BAERLIN2014 campaign (Berlin air quality and ecosystem research: local and long-range impact of anthropogenic and natural hydrocarbons) in Berlin, Germany (Bonn et al., 2016).

A frequently used approach to determine MLH is the implementation of so called ceilometers, automated and eye-safe single-wavelength backscatter lidars (Wiegner et al., 2014). As there is no strict definition of the technical specifications of a "ceilometer" recently the term "ALC" (automated lidars and ceilometers) has been introduced and is often used synonymously. Though originally designed for only determining cloud base heights ceilometers are now used for a variety of more sophisticated activities such as the retrieval of the particle backscatter coefficient $\beta_p$ and mixing layer height. Since ceilometers are commercially available including software providing "atmospheric products" (e.g. the MLH) we feel that it is necessary to scrutinize the application of such products. This is the main motivation and objective of our paper: to investigate the potential of proprietary software to derive the MLH, and the usefulness of correlations between such derived MLHs and surface concentrations of pollutants in an urban environment. The motivation of the latter is to increase the awareness that such correlations might be prone to over-interpretation. A thorough discussion of the meteorological reasons and atmospheric chemistry responsible for the observed distribution of pollutants is beyond the goal of the study.

For the determination of the MLH range corrected signals of ALC can be analyzed (e.g. Morille et al., 2007), often using proprietary software (e.g. Haman et al., 2012). Recently ceilometer networks have been installed by several national weather services (e.g. almost 100 instruments by the German Weather Service), and it is expected that in near future dense networks providing data in real time will be available on a European scale. Prospectively also the implementation of urban networks for air quality studies is likely at least for selected cities occasionally suffering from pollution events – recently three ceilometers were set up in larger Paris for this purpose (OCAPI: Observation de la Composition Atmosphérique Parisienne de l'IPSL).

However, the retrieval of MLH is an issue even though state-of-the-art ceilometers provide a clear identification of aerosol layers; often several atmospheric layers are detected but it remains ambiguous which one is the mixing layer. This problem can be severe, especially in case of automatic retrievals optimized for specific atmospheric conditions. Retrievals might fail or lead to under- or overestimates if the aerosol concentration is extremely low or high, or if the range of incomplete overlap of the instrument is too large. Consequently any correlation between MLH and pollution – and thus the potential to use the MLH in discussions of air quality – might depend on the selected MLH retrieval technique. In this paper we want to investigate this topic by applying different MLH retrievals provided by the manufacturer of the ceilometer (in our case Vaisala) and a novel scheme COBOLT (Geiß, 2016). We have calculated correlations with concentrations of pollutants at different locations in the metropolitan area of Berlin to compare the effects due to the spatial inhomogeneity of pollutants and due to uncertainties of the MLH

retrieval. The results may help to interpret possible links between air quality and MLH, even though there was only one ceilometer available during BAERLIN2014.

A selection of studies dealing with the link between MLH and pollutants is introduced in the next section. Then, we briefly describe the air quality network of Berlin and the measurement campaign. Section 4 provides a detailed description of different options to retrieve the MLH including a comparison. Correlations with concentrations of pollutants are discussed in view of their dependence on the selected MLH retrieval, and their location inside Berlin. A summary concludes the paper.

## 2   Relation between mixing layer height and surface concentrations

In this section a brief overview of studies dealing with the retrieval of the mixing layer height and its role with respect to air quality is given.

When discussing retrievals of the MLH it is important to note that it is defined in different ways, depending on the availability of specific measurement techniques and data sets. Most approaches are based on the analysis of either the temperature profile (e.g. Liu and Liang, 2010), the wind field (e.g. Schween et al., 2014) or concentration profiles of particles (e.g. Haeffelin et al., 2012). With the establishment of active remote sensing networks (e.g. the above mentioned ceilometer networks) the latter approach is gaining importance, basically it is assumed that the concentration of particles considerably decreases at the transition from the mixing layer to the free troposphere. Thus, the analysis of particle backscatter is a promising approach to determine the MLH.

A thorough review of approaches to determine the MLH was given by Seibert et al. (2000). They emphasized the benefit of active remote sensing techniques as they allow measurements of the vertical distribution of particles. Intercomparisons have shown (e.g. Emeis et al., 2004; Wiegner et al., 2006; Emeis et al., 2012) that sodar and RASS can be used to monitor MLH, however, they usually cannot provide the full diurnal cycle of the MLH in Central Europe, especially in summer (Piringer et al., 2007). Moreover, these techniques are less frequently applied mainly because of their more complicated implementation and higher expenses for investment and maintenance. The same is true for sophisticated multi-wavelength lidars (e.g. Baars et al., 2008), sodars (e.g. Beyrich, 1995), combinations of instruments (e.g. Cohn and Angevine, 2000), and combinations of models and measurements (e.g. Bachtiar et al., 2014). A large number of studies relying on lidar data has been published introducing different methodologies to determine MLH: among others Endlich et al. (1979) and Flamant et al. (1997) used algorithms based on first derivatives of the backscatter signal, Menut et al. (1999) used second derivatives, Hooper and Eloranta (1986) the temporal variance, Cohn and Angevine (2000), Brooks (2003) and Baars et al. (2008) applied wavelet covariance transforms, de Bruine et al. (2017) used graph theory, Caicedo et al. (2017) cluster analysis, and statistical methods were used by (e.g. Eresmaa et al., 2006; Lange

et al., 2014). With recent upgrades of the hardware those methodologies can also be applied to ALC, and with the implementation of networks they have become more attractive as they provide continuous monitoring and good spatial coverage.

The role of the mixing layer for pollution and its adverse effects on health have been highlighted since more than 50 years (Holzworth, 1964; Barlow, 2014). Consequently the link between air quality (in terms of particulate matter and concentrations) and MLH was investigated in many studies, primarily for urban areas. It should be emphasized that a comparison of different studies is inherently difficult, especially when only qualitative conclusions have been made. On the one hand different meteorological conditions and species are investigated, i.e. different gaseous pollutants and different sites in rural or urban environments. On the other hand there are conceptual differences, i.e. statistical analyses are based on hourly values, daily averages or diurnal cycles averaged over several weeks or even seasons, and there are different approaches to determine the MLH from measurements or numerical models. Moreover, there are differences with respect to the selection of a suitable MLH parameter used for correlation analysis: averages, medians or certain percentiles are used, maximum values, or MLHs are grouped in intervals.

Studies relying on numerical parameterizations were conducted e.g. by Tiwari et al. (2014) who used reanalysis data, and Du et al. (2013) who used routine meteorological observations to find an anti-correlation between $PM_{2.5}$ and MLH for Delhi, India, and Xi'an, China. Rost et al. (2009) found a strong anti-correlation between $PM_{10}$ and MLH (derived from radio sonde data) for Stuttgart, Germany, with a coefficient of determination of $R^2 > 0.95$. The awareness of the potential of active remote sensing started at the end of the last century when the first generation of ceilometers was deployed. These systems suffered from low pulse energies so that their use was confined to winter measurements or clear atmospheric conditions when the measurement range of the instrument was sufficient to cover the complete vertical extent of the mixing layer. Schäfer et al. (2006) deployed CT25k- and LD40-ceilometers (Vaisala) but primarily relied on co-located sodar-data when they found a high anti-correlation between $PM_{10}$ and MLH in Hannover and greater Munich, Germany, for winter conditions. They also found negative correlations for CO and $NO_x$ with quite variable $R^2$ depending on the site and the horizontal wind. Differences between summer and winter measurements were also observed. These findings agree with results from a campaign in Budapest, Hungary, with a similar set of instruments (Alföldi et al., 2007). Examples with state-of-the-art ceilometers include Beijing, China (e.g. Sun et al., 2013; Tang et al., 2015), Essen, Germany (Wagner and Schäfer, 2015), and Paris, France (Pal and Haeffelin, 2015), or rural sites (Pal et al., 2014). Some of these studies also investigated correlations with gaseous pollutants, e.g. Czader et al. (2013) for Houston, Texas. Significant negative correlations between surface NO concentration and MLH were reported for Beijing (Schäfer et al., 2012). However, surface $NO_2$ concentrations are only weakly affected by the MLH as they are mainly secondarily formed through

atmospheric processes. For Paris Dieudonné et al. (2013) investigated the relationship between surface concentrations of $NO_2$, column amount of $NO_2$ and the MLH. Their results suggest that the discrepancies between $NO_2$ surface concentrations and column amount can be explained by the differences in the MLH. For seven cities in the North China Plain an anti-correlation between near-surface $O_3$ and MLH was found (Hu et al., 2014) however, this case study was confined to only one day. Wagner and Schäfer (2015) investigated conditions near a major traffic road in Essen, Germany, and found that correlations between several constituents and MLH are significantly negative, if the MLH from the ceilometer measurements is grouped into intervals of 200 m.

Currently such investigations cannot ultimately demonstrate which correlations between surface concentrations and atmospheric stratification exist, how robust they are and how large their range of applicability is. One prerequisite for progress is a critical review of standard methods for the determination of the MLH. Then, the dependence of such correlations on season, meteorological conditions, or location can be investigated.

## 3   The BLUME network and the BAERLIN2014 campaign

Berlin is the capital of Germany with about 3,500,000 citizens. The terrain is flat with altitude differences of not more than 25 m except some small hills of up to about 85 m. A considerable part (about 40%) of the area of Berlin is covered by forests, agricultural areas, lakes and rivers. Similar to many other metropolitan areas Berlin suffers from episodes of poor air quality, in particular when particulate matter ($PM_{10}$) and $NO_2$ concentrations exceed the EU limit values. Thus, measures have been implemented such as restrictions of the traffic in the city center. Air pollution in Berlin originates not only from anthropogenic emissions of urban sources, but also from long range transport of particulate matter from industrialized areas in Poland, biogenic emissions and formation of secondary aerosol compounds; their relative contributions are not yet agreed upon in detail (Bonn et al., 2016).

Routine measurements of the air quality of Berlin are conducted at 16 automated stations of the so called BLUME-network (SenStadtUm (2014), see Fig. 1) under the responsibility of the Senate of Berlin by European law. Their main purpose is the monitoring of surface concentrations of trace gases and particle mass concentration. For this study hourly data are available. BLUME distinguishes three categories of stations: five of the stations are located at residential districts (labeled "urban background", grey flags), five at the outskirts of Berlin and forest areas ("rural background", green flags), and six at traffic hot spots (red flags). These data are reported to the Federal Environment Agency (UBA) of Germany and included into the European air quality database (AIR BASE).

A summary of the automatic stations of the BLUME network and the monitored quantities used in this study are given in Table 1. Particulate matter is measured with the automatic PMI (particulate monitoring

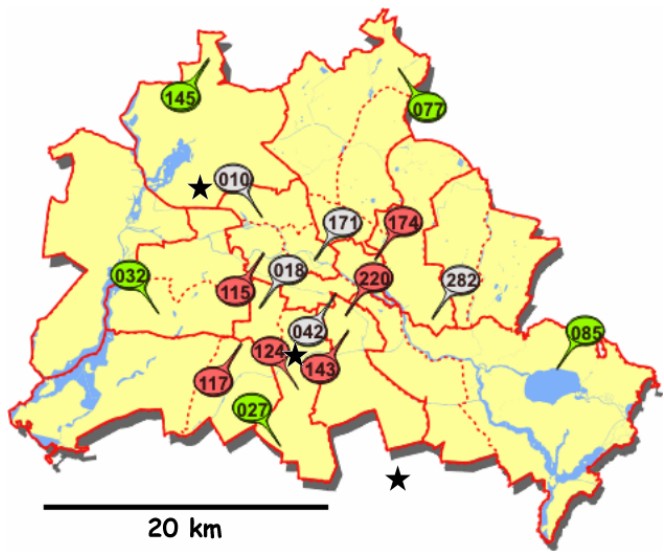

**Figure 1.** The 16 automated air quality station of the BLUME network. Ceilometer measurements reported in this paper were conducted at station #42, one of the urban background sites (in grey). Traffic stations are shown in red. The green flags are stations at the outskirts of Berlin and in forests (rural background). More details are given in Table 1. The three black stars indicate the wind measurements in Tegel, Tempelhof and Schönefeld (from northwest to southeast) mentioned in Sect. 5.1. Adaptation of a figure of the Senate of Berlin.

instrument), type FH 62 I-R (Thermo ESM Anderson), one of the standard systems for Germany's air quality network. It is based on attenuation of beta-radiation from a Krypton gas cell. It performs real time measurement of the suspended particulate matter on a filter. For the gaseous species discussed in this study Horiba's air pollution monitors (370-series) are deployed, i.e. an APNA-370 (for NO, $NO_2$

and $NO_x$, chemiluminescence method) and an APOA-370 (for ozone, absorption in the UV spectral range).

During summer 2014 a dedicated field campaign, BAERLIN2014 was set up for three months (from 2. June until 29. August 2014) deploying several additional measurements from mobile and airborne platforms focusing on ozone, secondary organic aerosols and the effect of urban vegetation (Bonn et

al., 2016). One Vaisala ceilometer was available at that time. It was installed at the BLUME station #42 (Nansenstraße, at the corner of Framstraße, 52.4894° N, 13.4309° E, see Fig. 1) on the roof of a children care take house (5 m above street level). This station is located in a residential neighborhood with trees and bushes. It is categorized as an "urban background" site: in 2014 annual averages were 27 $\mu$g/m$^3$ and 21 $\mu$g/m$^3$ of $PM_{10}$ and $PM_{2.5}$, respectively, 41 $\mu$g/m$^3$ of $O_3$, and 37 $\mu$g/m$^3$ of $NO_x$. The

$PM_{10}$-threshold (daily average of 50 $\mu$g/m$^3$) was exceeded on 28 days which is below the limit of 35 days according to EU-regulations (http://ec.europa.eu/environment/air/quality/standards.htm).

**Table 1.** Automatic stations of the BLUME network of Berlin: Names of the locations with the corresponding district are given in brackets. Coordinates are given as latitude (degree North) and longitude (degree East), $d_{42}$ is the distance [in km] from station #42 (Nansenstraße). Listed are only measurements of pollutants discussed in this paper.

| ID | location | coordinates | $d_{42}$ | | pollutants | |
|----|----------|-------------|----------|----|----|----|
| outskirts (rural background) | | | | | | |
| 27 | Schichauweg (Marienfelde) | 52.3984°, 13.3681° | 11.0 | | $NO_x$ | $O_3$ |
| 32 | Jagen (Grunewald) | 52.4732°, 13.2251° | 14.0 | $PM_{10}$ | $NO_x$ | $O_3$ |
| 77 | Wiltbergstr. (Buch) | 52.6435°, 13.4895° | 17.6 | $PM_{10}$ | $NO_x$ | $O_3$ |
| 85 | Müggelseedamm (Friedrichshagen) | 52.4477°, 13.6471° | 15.4 | $PM_{10}$ | $NO_x$ | $O_3$ |
| 145 | Jägerstieg 1 (Frohnau) | 52.6533°, 13.2961° | 20.3 | | $NO_x$ | $O_3$ |
| urban background | | | | | | |
| 10 | Amrumer Str. (Wedding) | 52.5430°, 13.3491° | 8.2 | $PM_{10}$ | $NO_x$ | $O_3$ |
| 18 | Belziger Str. (Schöneberg) | 52.4858°, 13.3488° | 5.6 | | $NO_x$ | |
| 42 | Nansenstr. (Neukölln) | 52.4894°, 13.4309° | 0 | $PM_{10}$ | $NO_x$ | $O_3$ |
| 171 | Brückenstr. (Mitte) | 52.5136°, 13.4188° | 2.8 | $PM_{10}$ | $NO_x$ | |
| 282 | Rheingoldstr. (Karlshorst) | 52.4853°, 13.5295° | 6.7 | | $NO_x$ | |
| traffic | | | | | | |
| 115 | Hardenbergplatz (Charlottenburg) | 52.5066°, 13.3330° | 6.9 | | $NO_x$ | |
| 117 | Schildhornstr. (Steglitz) | 52.4636°, 13.3183° | 8.2 | $PM_{10}$ | $NO_x$ | |
| 124 | Mariendorfer Damm (Mariendorf) | 52.4381°, 13.3877° | 6.4 | $PM_{10}$ | $NO_x$ | |
| 143 | Silbersteinstr. (Neukölln) | 52.4675°, 13.4417° | 2.5 | $PM_{10}$ | $NO_x$ | |
| 174 | Frankfurter Allee (Friedrichshain) | 52.5141°, 13.4699° | 3.8 | $PM_{10}$ | $NO_x$ | |
| 220 | Karl-Marx-Str. (Neukölln) | 52.4817°, 13.4340° | 0.9 | $PM_{10}$ | $NO_x$ | |

The objective of the ceilometer measurements was the determination of the MLH and thus the option to combine in-situ measurements at the surface with data concerning the vertical direction. Based on previous case studies for Munich (Wiegner et al., 2006) and Paris (Pal et al., 2012), long term observations for the region Munich/Augsburg/Freising (Geiß, 2016) and for Vienna (Lotteraner and Piringer, 2016) we assume that the so derived MLH is representative for Berlin. As can be seen from Table 1 all sites are within 20 km distance from the ceilometer with five stations being very close (less than 6.4 km).

Note, that all times are given in CET (central European time).

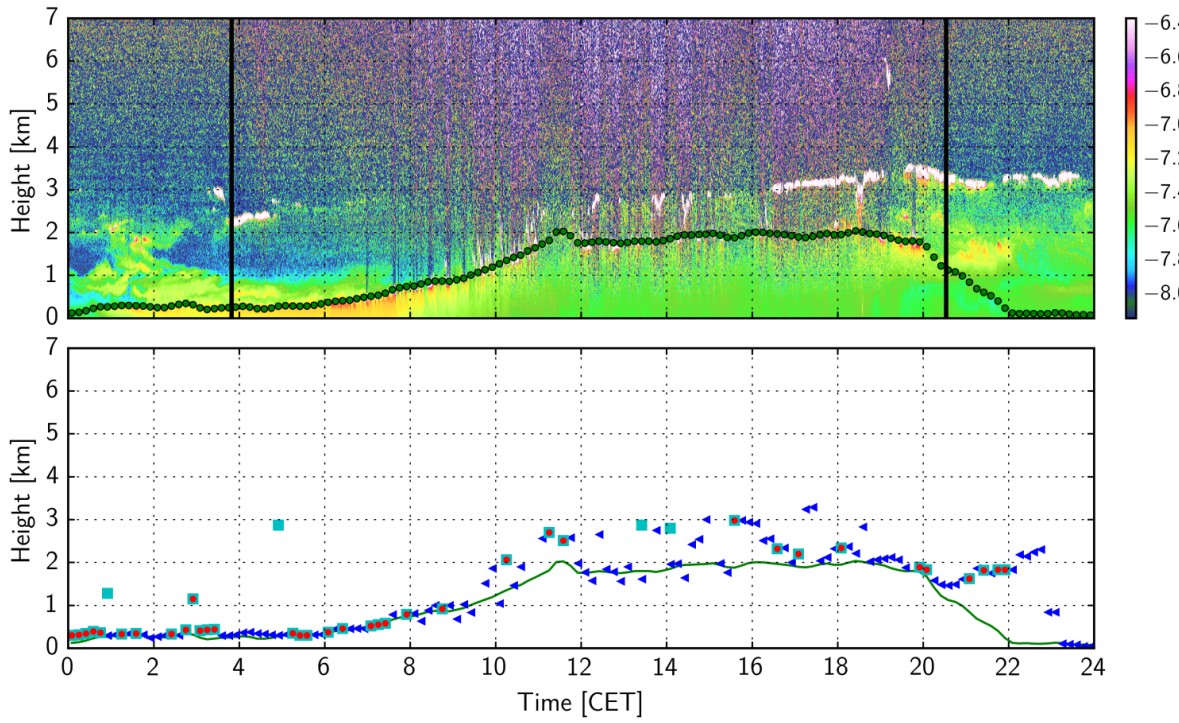

**Figure 2.** Top: Time-height cross section of range corrected ceilometer signals (Vaisala CL51) at the BLUME network site #42, Nansenstraße, on 1. July 2014 (in arbitrary units). The MLH as determined from COBOLT ($H_{ml,c}$) is marked by dark green dots. Bottom: Comparison of the MLH retrievals: $H_{ml,c}$ as above (green line), $H_{ml,v}$ with L1-criterion (blue triangles), the L3-criterion (red dots), and the Q3-criterion (cyan squares); for the definition see Table 2.

## 4 Mixing layer heights from ceilometer measurements

### 4.1 Ceilometer data

In the framework of BAERLIN2014 a Vaisala CL51 ceilometer (Münkel, 2007) was deployed. The instrument is fully automated and eye-safe. It provides backscatter signals at 910 nm. As this wavelength is influenced by water vapor absorption it is complicated to derive optical properties of particles in a quantitative way (Wiegner and Gasteiger, 2015), however the identification of aerosol layers is not affected as strong changes of the aerosol backscatter are not masked by the water vapor absorption. The height range of the measurements is more than 4 km, thus covering the typical range of MLH over a continental site like Berlin. Due to its optical design using the same lens for the emitter and the receiver optical paths, the minimum range is on the order of 50 m for the detection of aerosol layers and even lower for clouds. The spatial and temporal resolution is 10 m and 16 s, respectively. Ceilometer data

(firmware version V1.032) are available for 67 days between 27. June and 2. September 2014 (except 15. July). The output signals are range corrected consistently for the whole measurement range, i.e., the "H2on"-parameter was set to 1 as discussed by Kotthaus et al. (2016). To improve the detection of aerosol layers close to the ground, an additional overlap correction function, similar to a concept

outlined by Hervo et al. (2016), was applied.

## 4.2    Determination of the MLH

Virtually all retrievals of the MLH from ceilometer measurements are based on the shape of the range corrected signal (i.e. uncalibrated) or the vertical profile of the attenuated backscatter coefficient (i.e. calibrated, Wiegner and Geiß (2012)). Several methods to analyze the gradient of the profile or its

temporal variability are available, different thresholds can be selected to distinguish between clouds and aerosol layers, and different temporal and vertical averaging can be applied to reduce the influence of noise.

The standard procedure for the MLH determination from Vaisala-ceilometers ($H_{ml,v}$) is the MATLAB-based software package BL-VIEW developed by the manufacturer. It provides up to three

altitudes of aerosol layers (referred to as candidate levels in the following); they are counted upward, i.e., candidate level #1 is closest to the ground. They are determined from local minima of the gradient of the backscatter profile considering data of a 14 minutes-time period prior to the actual measurement; in case of low signal-to-noise ratios this time span is extended to 20 minutes. To improve the retrieval, signals are smoothed along the line of sight, thresholds are defined to identify cloud "contamination",

and unrealistic outliers are deleted. In case of rain, no $H_{ml,v}$ is provided. Each candidate level is given with a quality flag based on the absolute value of the gradient and the "width" of the local minimum (Münkel et al., 2011). Quality flags are 1, 2 or 3, with 3 meaning the highest reliability. Candidate levels with quality flag 3 are not necessarily given for all times. This information is stored in an ASCII-file, and it is left to the user to find their own criteria to determine the MLH, i.e., different selection of the

candidate levels is possible, and the quality flags might be considered or not. The advantage of the provision of three candidate levels is that different layers can be detected at the same time (e.g., stable layer, convective layer, residual layer), the disadvantage is that the attribution of the layers is more complicated (Schween et al., 2014). The details of BL-VIEW are not disclosed to the user.

In this paper we use different criteria. To facilitate further reading we introduce the acronym "L1" for

the criterion "lowest candidate level if it has a quality flag of at least 1" (this is identical to the condition "lowest candidate level without considering the quality flag"'). "L2" and "L3" are defined accordingly. So in all cases the lowest candidate level (#1) is chosen if the quality flag fulfils the corresponding conditions, otherwise no MLH is retrieved. "Q3" stands for the criterion "lowermost candidate level with quality flag 3", meaning that any candidate level is chosen as long as it has the best quality flag.

**Table 2.** Overview over different approaches to determine MLH from BL-VIEW: the conditions with respect to the quality flag and the number of the candidate level

| acronym | selected candidate level | quality flag |
|---------|--------------------------|--------------|
| L1 | #1 | 1, 2 or 3 |
| L2 | #1 | 2 or 3 |
| L3 | #1 | 3 |
| Q3 | lowermost of #1, #2 or #3 | 3 |

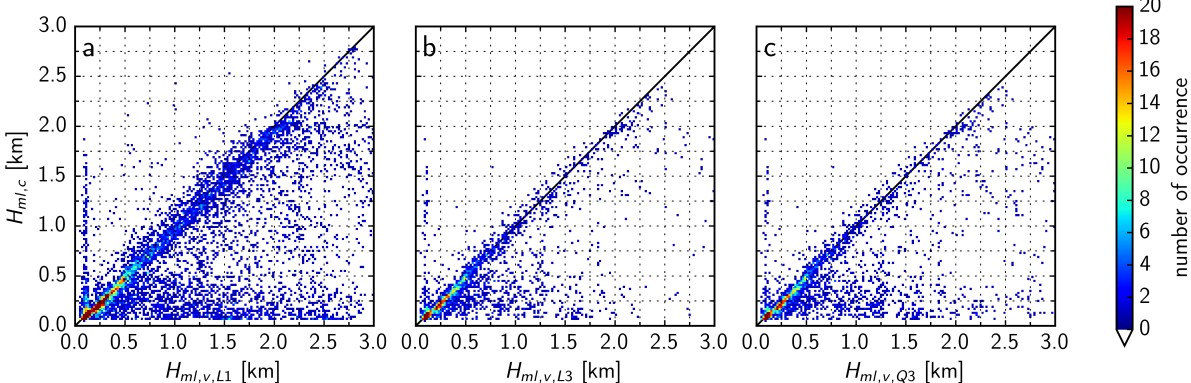

**Figure 3.** Comparison of MLH [in km] retrieved by COBOLT ($H_{ml,c}$) and different BL-VIEW approaches during the BAERLIN2014 campaign. (a): $H_{ml,v,L1}$ from L1-criterion, (b): ($H_{ml,v,L3}$ from L3-criterion, and (c): $H_{ml,v,Q3}$ from Q3-criterion. The number of occurrence is color coded.

If more than one candidate level fulfils this quality criterion, the lowermost is selected. For reasons of clarity our nomenclature is summarized in Table 2. It is obvious that L1 is more often fulfilled than L3, and that any successful retrieval according to L3 is also a successful Q3-retrieval.

An alternative approach to determine the MLH ($H_{ml,c}$) has been developed by Geiß (2016), referred to as COBOLT ("continuous boundary layer tracing"). The code is written in the open source programming language Python and can be used on Windows and Linux platforms. The algorithm is based on a time and height dependent function $A(t,z)$ that has been defined according to Eq. 1:

$$A(t,z) = \frac{\epsilon_g\, M_g(t,z)}{99^{\text{th}}(\epsilon_g\, M_g(t,z))} + \frac{\epsilon_v\, M_v(t,z)}{99^{\text{th}}(\epsilon_v\, M_v(t,z))} \tag{1}$$

It depends on the magnitude and orientation of gradients of the range corrected ceilometer signal (first term on the right hand side), and on the temporal variability of the aerosol layering (second term). Both terms are weighted according to $\epsilon_g$ and $\epsilon_v$, respectively, and are normalized by the 99. percentile

of the function. By applying the Sobel operator (Duda and Hart, 1973), in principle a two-dimensional gradient method, to $X_\Lambda$,

$$X_\Lambda(t,z,a,b) = \frac{1}{a} \int\limits_{z_0}^{z_{max}} X(t,z)\Lambda\left(\frac{z-b}{a}\right) dz, \tag{2}$$

with $X(t,z)$ as range corrected ceilometer signal and a low-pass filter $\Lambda\left(\frac{z-b}{a}\right)$ defined as

$$\Lambda\left(\frac{z-b}{a}\right) = \begin{cases} \frac{a}{2} - z + b & \text{if} \quad b - \frac{a}{2} \leq z \leq b \\ \frac{a}{2} + z - b & \text{if} \quad b \leq z \leq b + \frac{a}{2} \\ 0 & \text{elsewhere.} \end{cases} \tag{3}$$

the function $M_g(t,z)$ and the direction of the gradients $\Theta(t,z)$ are obtained. The application of the Sobel operators to a low pass filtered ceilometer signal is equivalent to the wavelet covariance transform method using a Haar wavelet (Comeron et al., 2013). Parameters $a$ and $b$ in Eq. 3 are the wavelet dilation and translation, respectively. The advantage of the Sobel operator is that both temporal and spatial changes can be evaluated simultaneously. The weighting function $\epsilon_g(t,z)$ is defined such that
MLH that are unlikely in a meteorological sense are suppressed:

$$\epsilon_g(t,z) = \begin{cases} 0.1 & \text{if} \quad 0° \leq \Theta \leq 185° \\ 0.1 & \text{if} \quad 355° \leq \Theta \leq 360° \\ 1 & \text{elsewhere} \end{cases} \tag{4}$$

With this definition e.g. range corrected ceilometer signals that increase with height ($\Theta \approx 90°$) – and most likely do not represent the top of the mixing layer – have a very low weight. In contrast, negative
gradients caused by decreasing aerosol backscattering with height ($\Theta \approx 270°$) are emphasized. $M_v(t,z)$ is the temporal variance of $X_\Lambda(t,z)$ and the weighting factor $\epsilon_v(t,z)$ is height-dependent in order to account for the decreasing signal to noise ratio with height. Specific gradient angles are excluded:

$$\epsilon_v(t,z) = \begin{cases} 0 & \text{if} \quad -5° \leq \Theta \leq 5° \\ 0 & \text{if} \quad 175° \leq \Theta \leq 185° \\ 1 - \frac{z}{3\,\text{km}} & \text{elsewhere, } z \text{ in km} \end{cases} \tag{5}$$

The function $A(t,z)$ was defined to especially determine the height of the convective boundary layer.
The empirical weights $\epsilon_g$ and $\epsilon_v$ had undergone extensive testing to find solutions that provide a reliable identification of the top of the mixing layer from the maximum of $A(t,z)$. For the determination of the

diurnal cycle of the MLH, the maximum of $A(t,z)$ is traced in time. For the initialization of the time-height tracking procedure $H_{ml,c}$ at a starting time $t_0$ is required. It is determined between 2.5 hours and 3.5 hours after sunrise, when the convective mixing layer is assumed to be existent (Wildmann, 2015). Relying on the variance method which is especially sensitive to the beginning convection (Menut

et al., 1999), the height of the maximum value of $A(t,z)$ is chosen as the initial $H_{ml,c}(t_0)$. Starting with $H_{ml,c}(t_0)$ a search window with a vertical extent depending on the solar zenith angle is moved backward in time to cover the period before sunrise, and forward until sunset. In case of rain $H_{ml,c}$ remains unchanged but is flagged; consequently, observations during (long lasting) precipitation events can be excluded by the user if desired. In the presence of convective clouds at the top of the boundary

layer, the strongest decrease of the signal in the cloud is used to determine $H_{ml,c}$, which is usually a few tens of meters above the cloud bottom. The analogue procedure as for the convective daytime MLH is applied after sunset for the nocturnal stable boundary layer. To account for the transition from decaying thermals in the well developed mixing layer to the establishment of a stable boundary layer a linear change of the $H_{ml,c}$ between both layers is assumed to take place between 30 minutes before until 60

minutes after sunset (Grant, 1997; Grimsdell and Angevine, 2002).

In COBOLT an ensemble of 40 potential tracks $H_{ml,c}(t)$ is calculated with different initial conditions and search criteria, e.g. different widths of the search window. The selection of the final result is performed by means of the function $C_j$ for each ensemble member $j$ ($\leq 40$) as defined in Eq. 6

$$C_j = \frac{\sum_{i=0}^{N-1} \sqrt{(t_{i+1}-t_i)^2 + (H_{ml,c}(t_{i+1}) - H_{ml,c}(t_i))^2}}{\sum_{i=0}^{N} A(t_i, H_{ml,c})} \tag{6}$$

with $N$ being the number of time steps $t_i$ within one day, i.e. $N$=144 for COBOLT's temporal resolution of 10 minutes. The track $j$ with the minimum value of $C_j$ is selected as the final result: the main idea behind this selection is that the MLH is assumed to develop smoothly in time, i.e., sudden "jumps" (that would increase the length of the track) do not occur in reality but are caused by wrong attribution of the mixing layer in case of multi-layered aerosol distributions. As a consequence, COBOLT retrievals

do not have any temporal gaps, and unrealistic growth rates of $H_{ml,c}$ are suppressed. Otherwise, in particular in case of the detection of e.g. two layers it might happen, that the retrieved $H_{ml,c}$ "switches" between those layers resulting in very strong and rapid changes.

To make both approaches better comparable, time is assigned to the center of the interval of the BL-VIEW retrieval. Note, that a perfect temporal co-incidence is not possible because of the inherent

properties of both algorithms, e.g. the height-dependent temporal averaging in case of COBOLT.

### 4.3 Comparison of MLH retrievals

A typical example of CL51 measurements and the MLH retrieval is shown in Fig. 2. The attenuated backscatter signal (color-coded, in arbitrary units) up to 7 km above ground is shown in the upper panel for 1. July 2014. Sunrise at 03:46 CET and sunset at 20:32 CET are highlighted by the black lines. Visual

inspection shows broken cloud fields from 09:00 CET to 16:00 CET at different altitudes, afterwards an almost continuous cloud deck at 3 km, and inhomogeneous aerosol layers up to 2.0 km before sunrise and up to 3.0 km after sunset. The MLH as identified by COBOLT ($H_{ml,c}$) is marked by dark green dots.

    The results of all MLH retrievals are shown in a separate panel for reasons of clarity (Fig. 2 bottom):

BL-VIEW ($H_{ml,v}$) with different selection criteria L1, L3 and Q3 are shown as blue triangles, red dots, and cyan squares, respectively, whereas $H_{ml,c}$ is shown as green line (same as in the upper panel). The temporal interval is 10 minutes. It can be seen that the overall agreement between COBOLT and BL-VIEW L3 is very good and coincides with what a human observer would have analyzed. Note that in general cloud bottoms were not misinterpreted as MLH by either approach. For L2 (not shown here) and

L1 more cases of wrong assignments occur. Disagreements between COBOLT and L3 are rare, mainly between 20:00 CET and 22:00 CET when $H_{ml,v}$ is significantly higher – here the residual layer seems to be interpreted as the mixing layer by the Vaisala retrieval. Disagreements are more frequent when L1 or L2 is applied instead of L3, e.g. around noon, when BL-VIEW L1 selects the top of elevated aerosol layers and occasionally clouds as the MLH, or after sunset, when L1 selects the residual layer. It is

obvious, that $H_{ml,v}$ is often not available during the daylight period, especially when L3 is considered. The main reason is the high temporal variability of the distribution of aerosol particles and clouds, e.g. under not well-mixed conditions with more than one aerosol layer that prohibit an unambiguous determination of $H_{ml,v}$. Consequently, candidate levels are rapidly changing, leading to lower quality flags (Münkel et al., 2011) and a failure of the MLH assessment. So it can be understood that the

temporal coverage of $H_{ml,v}$ is quite low if L3 is applied. Figure 2 confirms that even the application of L1 (and L2, not shown here) does not fill all temporal gaps. As all MLHs from L3 are by definition also fulfilling the Q3-criterion, these results do not differ much. Only very few cases are added, e.g. before sunset, when the top of the residual layer was identified as the second or third candidate level and flagged with the highest quality.

These conclusions also hold for the whole period of BAERLIN2014. The intercomparison of the different MLH retrievals is summarized in Fig. 3. Figure 3a concerns BL-VIEW when the "weak" constraint L1 is applied: for each {$H_{ml,v,L1}$, $H_{ml,c}$}-pair the number of occurrence is color-coded. As expected from the example shown in Fig. 2, many cases with $H_{ml,v,L1} > H_{ml,c}$ exist. This is a consequence of multiple aerosol layers and the different behavior of the algorithms in the presence of a resid-

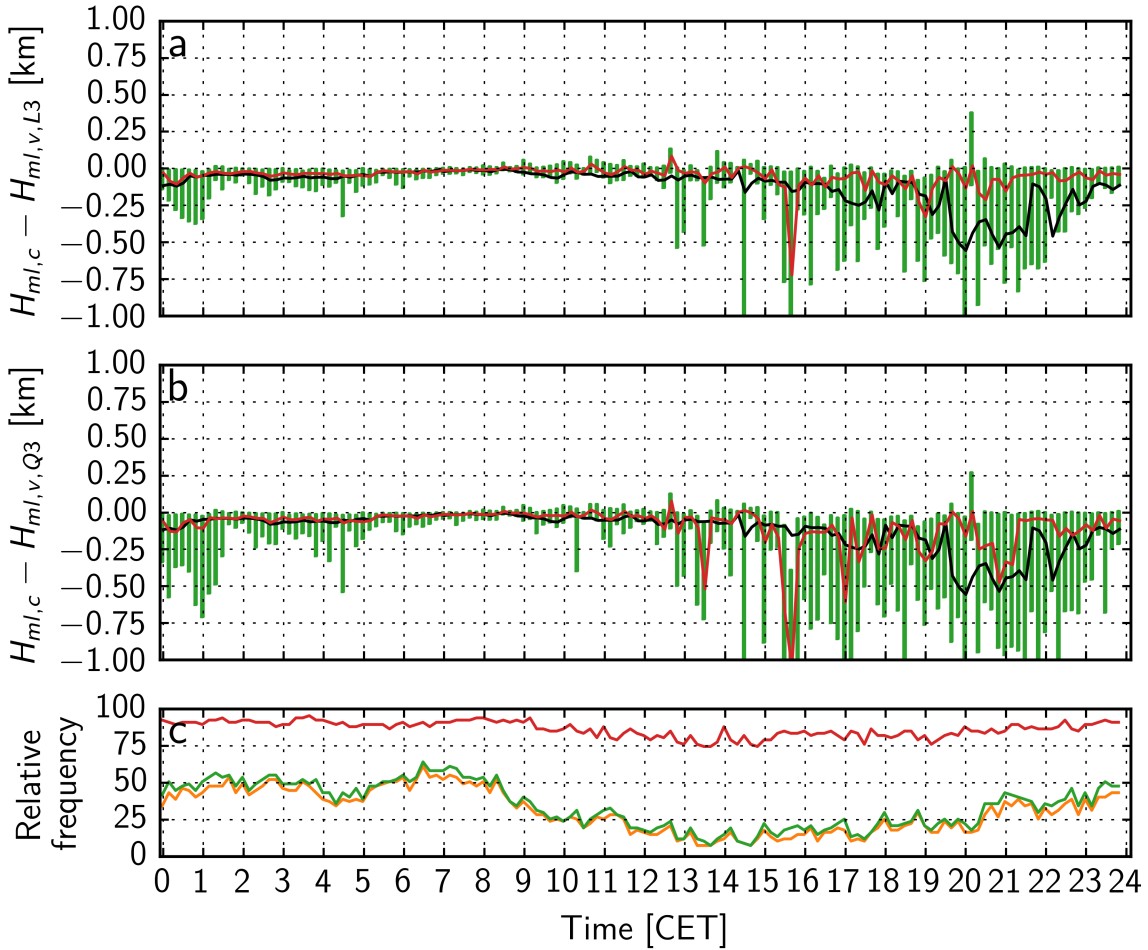

**Figure 4.** a: Difference $\Delta H$ of the retrieved MLH from COBOLT and BL-VIEW L3 during the BAERLIN2014 campaign: vertical lines indicate the interval from the 25th to the 75th percentile. The red line is the median of the distribution. For comparison the corresponding median from the L1-criterion is shown (black line). b: same as top panel but $\Delta H$ of the retrieved MLH from COBOLT and BL-VIEW Q3. c: relative frequency of successful $H_{ml,v}$-retrievals (L3 in orange, Q3 in green, L1 in red) in percent in relation to the COBOLT-retrieval.

ual layer. The correlation coefficient according to Pearson is $R = 0.653$. The corresponding comparison if the stronger constraint L3 is applied is shown in Fig. 3b. Here, the correlation is obviously better with $R = 0.754$. Again, the number of cases with $H_{ml,v,L3} > H_{ml,c}$ is much larger than the opposite case, but less frequent than before (Fig. 3a). Similar results are found when Q3 is applied (Fig. 3c). It is the same distribution as the L3-case however with some additional cases when the lowest candidate level has a low quality flag, whereas one of the upper levels is considered as quite reliable. Consequently, the additional points concern primarily large $H_{ml,v}$ and the correlation is lower than before ($R = 0.650$). It is clear, that the application of more rigorous criteria leads to a drastic reduction of successful $H_{ml,v}$-retrievals: with the L1-criterion the total number is 8346, whereas it is only 2998 and 3331 for L3 and Q3, respectively. Note, that the largest possible number of MLH retrievals would be 9648 ($67 \times 24 \times 6$).

To better understand the reasons for the discrepancies between the approaches the difference ($\Delta H$)

$$\Delta H(t) = H_{ml,c}(t) - H_{ml,v}(t) \tag{7}$$

for each 10 minutes interval is calculated. Figure 4a concerns the L3-criterion. Green bars show the range between the 25th and 75th percentiles of $\Delta H$ at a given time. The red line illustrates the median value. For comparison the corresponding median of the L1-approach (black line) is also shown. It is obvious that the median is very small for both BL-VIEW approaches and stays between $+0.03$ and $-0.11$ km before noon. Between 16:00 CET and 23:00 CET $\Delta H$ is clearly shifted to negative values with a median reaching $-0.33$ km and $-0.56$ km for L3 and L1, respectively. This is a clear indication that with the establishment of the residual layer in the late afternoon and after sunset, the BL-VIEW algorithm tends to select the top of the residual layer as $H_{ml,v}$, especially if the user selects the L1-criterion. A similar effect is found in cases of complex aerosol particle distributions with several layers. L3 gives a much better agreement with COBOLT, however, as already mentioned, the stricter L3-criterion leads to considerable temporal gaps in the $H_{ml,v}$-retrieval: in Fig. 4c it can be seen (orange line) that the relative number of 10 minutes intervals that allows to determine $H_{ml,v}$ is never larger than 61 %. Between 10:00 CET and 20:00 CET it is typically only in the 15–25 %-range because in the majority of cases the lowest candidate level does not have the highest quality flag (see Fig. 2). The low number of successful retrievals is also the reason for the rare cases (e.g., at 15:40 CET) when the absolute value of $\Delta H$ for L3 is larger than for L1. If the weaker L1-criterion is applied the availability of $H_{ml,v}$ is significantly increased (see the red line) and reaches a relative frequency of successful retrievals of more than 75 % throughout the day, however, at the expense of a in general good agreement between $H_{ml,v}$ and $H_{ml,c}$.

The corresponding comparison for the Q3-criterion is shown in Fig. 4b. The findings are similar as before, however, the range of differences $\Delta H(t)$ is extended towards larger negative values (green lines) as expected. This concerns the whole diurnal cycle but the effect is strongest after sunset. The number of successful $H_{ml,v}$-retrievals is slightly larger than for the L3-criterion as can be seen in the lower panel

(green line).

If we compare – as a consequence of these findings – only MLH retrievals before sunset the agreement between BL-VIEW and COBOLT is indeed improved. If the L1-criterion is applied to the complete diurnal cycle 23.4 % of the intercomparisons show large negative differences ($\Delta H < -0.5$ km). If only measurements before sunset are considered the number is reduced to 19.1 %. The corresponding

numbers for the Q3-criterion are 20.2 % and 17.3 %, respectively. For the L3-criterion we find 12.3 % and 9.5 %. Retrievals when $H_{ml,c}$ is larger than $H_{ml,v}$ are quite rare. A difference $\Delta H$ of more than 0.5 km occurs in less than 1.5 % of the cases for all three BL-VIEW approaches.

Figure 5 shows the mean diurnal cycle of MLH from 67 days as retrieved by BL-VIEW L3 and COBOLT. The dark blue line corresponds to $H_{ml,c}$ whereas the orange line is for $H_{ml,v}$. The mean

maximum vertical extent is approximately 1.5 km, similar to results from Lotteraner and Piringer (2016) found for Vienna. The light blue lines indicate the temporal variability as calculated from the standard deviation $\sigma_c$ (COBOLT approach). It is in the order of 100 m before sunrise and up to 500-700 m in the afternoon. Though this finding is based on COBOLT that provides complete temporal coverage it remains open whether this is representative for summer months in Berlin. Similar values but less

variability were found for Barcelona, Spain (Sicard et al., 2006). From summer observation over five years in Paris, France, Pal and Haeffelin (2015) found larger values ($H_{ml}$=1.95 $\pm$ 0.38 km), whereas maxima less than 0.8 km were observed during two years at Vancouver, BC, Canada (van der Kamp and McKendry, 2010) and Santiago, Chile (Munoz and Undurraga, 2010). The mean $H_{ml,c}$ at night is in the range of 0.2 km underlining the need of ceilometers with a very low overlap (or a reliable overlap

correction function, see e.g. Hervo et al. (2016) for a CHM15k-ceilometer) for investigations of the mixing layer. The most prominent differences between BL-VIEW and COBOLT are the larger $H_{ml,v}$ during night, and the rapid changes of $H_{ml,v}$ around noon. The main reason for these "fluctuations" is the low number of retrievals when L3 is applied, e.g. for some of the 10 minutes intervals only in 5 out of 67 days $H_{ml,v}$ could be found. Thus, the significance is limited, nevertheless $H_{ml,v}$ is within the

range of $H_{ml,c} \pm \sigma_c$.

The green line in Fig. 5 shows the first derivative of the COBOLT retrieval $H_{ml,c}$. This quantity can be relevant in view of temporal averaging, e.g., when MLH is correlated with concentration measurements having a lower temporal resolution. This topic is briefly discussed in the next section.

It is worthwhile to also determine a typical afternoon value of MLH. Figure 5 confirms that this period

provides the maximum volume for the mixing of emitted compounds, and that the MLH is representative

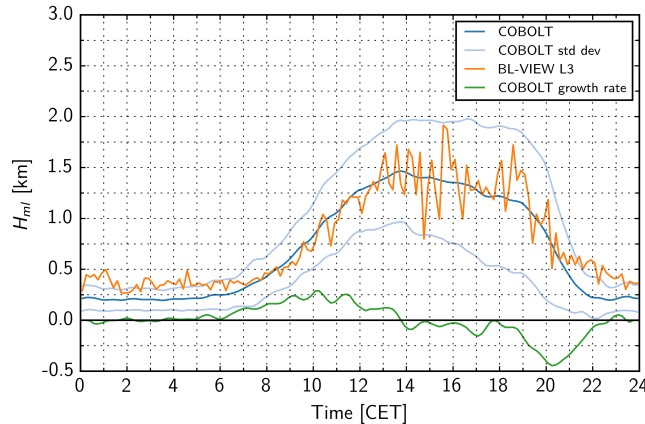

**Figure 5.** Mean diurnal cycle of $H_{ml,c}$ (dark blue) and $H_{ml,c} \pm \sigma_c$ as retrieved with COBOLT, and $H_{ml,v}$ from BL-VIEW L3 (orange) at the urban background site #42, in 10 minutes resolution, averaged over 67 days. The green line shows the growth rate of $H_{ml,c}$ (in km/h).

for several hours. The latter has been the reason for including a measurement around two hours after local noon in the regular EARLINET schedule (Pappalardo et al., 2014). Based on the mean diurnal cycle we define this value as the average over the 3-hour time period starting 30 minutes after noon. Figure 6 shows the results from COBOLT (blue dots) and L3 (orange dots) for the whole period of 5 67 days. Note, that BL-VIEW with the strict L3-criterion fails to determine $H_{ml,v}$ in 21 days (shaded areas) for the reasons mentioned above. If both values are available the general agreement is however good, only few cases exist when $H_{ml,v}$ is much larger than the respective COBOLT-result $H_{ml,c}$ (e.g., 27. June, 1. July, and 10. July).

We conclude that the main discrepancies between COBOLT and BL-VIEW origin from the presence 10 of the residual layer and elevated aerosol layers during day time whereas broken cloud fields cause less problems. The main drawback of the present version of BL-VIEW is the limited temporal coverage, when only retrievals with the highest quality flag are considered.

## 4.4 Temporal averaging of the mixing layer height

When evaluating ceilometer data a temporal resolution of MLH-retrievals of the order of 10 minutes can 15 be achieved. This is typically better than the resolution of air quality measurements of automated monitoring stations. To make MLH-retrievals comparable with the in-situ measurements of the BLUME-network, 1-hour averages have to be calculated. In this context the growth rate of the mixing layer $(dH_{ml}/dt)$ is relevant; it is shown for the mean diurnal cycle derived from COBOLT as a green line in Fig. 5. It can be seen that the mean $H_{ml,c}$ rises with $150 - 200$ m per hour between 08:00 and 12:00

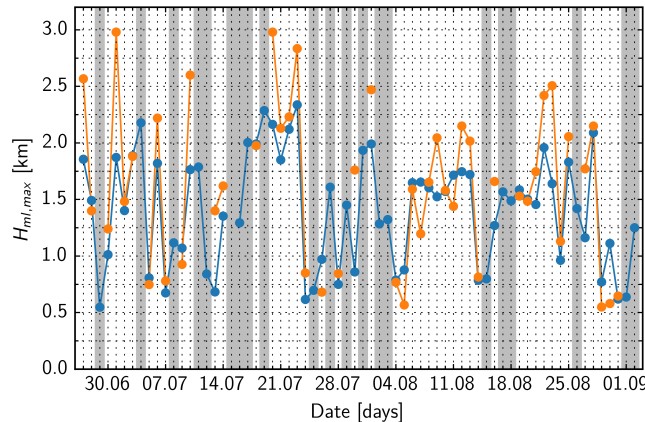

**Figure 6.** Daily afternoon value of $H_{ml}$ (averaged over 3 hours starting 30 minutes after noon) as derived from COBOLT (blue dots) and BL-VIEW L3 (orange dots) between 27. June and 2. September 2014. The alignment of the labels of the $x$-axis (date) is defined by the position of the dots separating day and month. The shaded areas highlight days when $H_{ml,v}$ could not be retrieved applying the L3-criterion.

CET with a maximum of 290 m. This is in good agreement with other continental cities (e.g. Baars et al., 2008; Pal and Haeffelin, 2015). The mean diurnal cycle of $H_{ml,c}$ shows its strongest decrease after sunset, reaching rates of $-450$ m per hour. For individual days these rates can be exceeded. However, in case of L3 or Q3 the MLH cannot be retrieved for each 10 minutes interval (see low values in Fig. 4c).

As a consequence, hourly averages of the MLH can be biased on the order of $\pm 100$ m due to the rapid growth of the mixing layer during strong convection events before noon. After sunset the uncertainty can be even larger ($\pm 200$ m).

Medians of the MLH are derived from all available 10-minutes retrievals (up to six, depending on the retrieval) of the corresponding hour, for all 67 days. So, up to 402 values are considered. The resulting

hourly values as they are used in the following discussion (Sect. 5) are shown in Fig. 7. In particular before sunrise averages are larger than the medians of MLHs. This is expected from Figs. 4a and 4b showing negative values of $\Delta H(t)$, i.e. there are cases of much larger MLH derived from the BL-VIEW retrievals.

## 5   Link to air quality

In the following we consider BLUME measurements of $PM_{10}$ and concentrations of $O_3$ and $NO_x$. These measurements are available with a temporal resolution of 1 hour. For the MLH we use the arithmetic mean of up to six values from 10 minutes intervals.

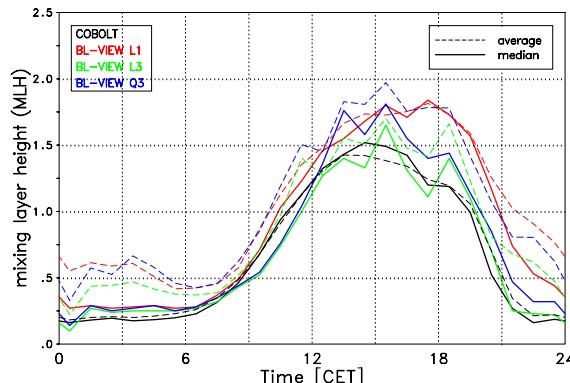

**Figure 7.** Diurnal cycle of hourly values of the MLH [in km] as determined from the different retrievals (see legend). Thick solid lines are for medians, thin broken lines for averages.

Episodic mobile (bicycle) measurements during BAERLIN2014 have already shown that there is significant horizontal heterogeneity in gas-phase pollutants and particle number concentrations (Bonn et al., 2016). In the following we discuss the influence of different retrievals of the MLH on correlations with surface measurements of $PM_{10}$ and concentrations of gases ($O_3$ and $NO_x$). Note, that in-situ mea-
surements are available at different sites, whereas only one ceilometer was deployed, consequently an inherent assumption of the following discussion is, that the MLH is the same over Berlin.

## 5.1   Correlation between MLH and $PM_{10}$

For the discussion of correlations between MLH and $PM_{10}$ we can use measurements at the outskirts (#32, #77, #85), at urban background stations (#10, #42, #171) and at five stations that are strongly
influenced by traffic (#117, #124, #143, #174, #220, see Fig. 1 and Table 1). The diurnal cycles of $PM_{10}$ (in $\mu g/m^3$) at these eleven stations are shown in Fig. 8, calculated as medians of all measurements of the corresponding hour of each day of the measurement period (67 days). It can be seen that the concentrations at the traffic sites (solid lines) are in general slightly higher than at the urban background and the outskirts. The amplitude of the mean diurnal cycle is quite small – between 4.4 $\mu g/m^3$ at #32
(red dotted line) and 10.6 $\mu g/m^3$ at #124 (green solid line) – whereas the day to day variations are comparably large at all sites and all times of a day. On the one hand the diurnal cycles have some common features; e.g., a distinct increase during the morning rush hours at all traffic sites and some of the urban background sites. This is plausible from vehicle emissions. At the urban background site #171 and sites at the outskirts, however, the strong increase occurs several hours later. The delay might be

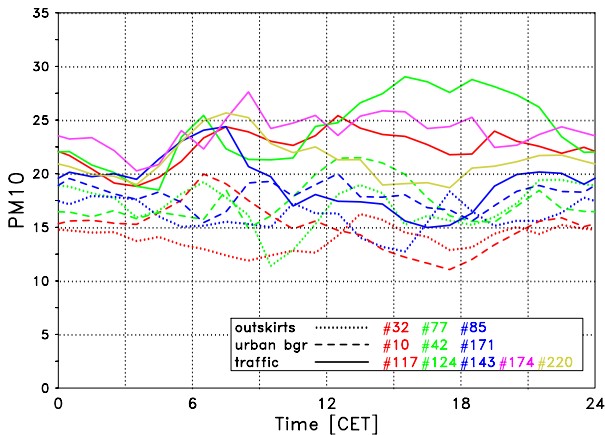

**Figure 8.** Diurnal cycle of $PM_{10}$ in $\mu g/m^3$ at eleven BLUME stations, based on medians of measurements on 67 days. The temporal resolution is 1 hour. The locations at the outskirts of Berlin (dotted lines), urban background sites (dashed) and traffic sites (solid) are indicated in the legend; see also Table 1.

caused by the transport time from the main sources to the site. On the other hand changing contributions of large scale transport from variable directions, local sources or particle removal by precipitation can lead to a quite different development in the course of a day including continuously increasing/decreasing $PM_{10}$, sporadic "peaks", or sudden drops at any time. The combination of these effects complicates a
meteorological interpretation of mean diurnal cycles.

For the determination of the diurnal cycle of the MLH we have – as already mentioned in Sect. 4.2 – four different MLH retrievals available. For the correlations between the $PM_{10}$-measurements and the MLH retrievals, further options can be considered: either averages or medians of hourly values (67 or less) as shown in Fig. 7 can be used. Figure 9 illustrates how these correlations depend on the site and
the retrieval. Eleven blocks according to the eleven sites are separated and labelled following Table 1. For each site four different correlations are shown (from left to right): averages of MLH vs. averages of $PM_{10}$ , medians of MLH vs. medians of $PM_{10}$, averages of MLH vs. median of $PM_{10}$, and median of MLH vs. averages of $PM_{10}$. The different colors indicate which ceilometer retrieval is used to determine the MLH: the COBOLT-approach is shown in black, and the BL-VIEW retrievals in red (L1-criterion),
green (L3) and blue (Q3).

The wide range of correlation coefficients for the different locations is obvious: The strongest correlation between MLH and $PM_{10}$ is found for the traffic site #124 ($R \approx 0.77$), the strongest anti-correlation for site #143 (traffic, $R \approx -0.79$) and site #10 (urban background, $R \approx -0.78$). So only for two sites a correlation is found as is expected if vertical dilution were the dominant process for the surface con-
centration of particulate matter. Compared to the large spatial heterogeneity the differences for different

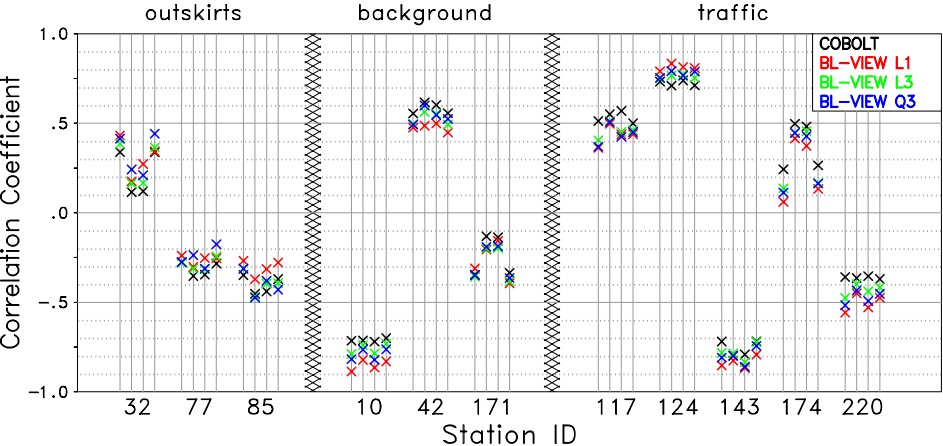

**Figure 9.** Correlation coefficient $R$ for mean diurnal cycles of MLH and $PM_{10}$ for eleven sites (from left to right): sites on the outskirts (#32, #77, and #85), urban background sites (#10, #42, and #171), and traffic sites (#117, #124, #143, #174, and #220). The results of the different retrievals are color coded as indicated in the legend. The four vertical lines of each block correspond to different options of correlation: MLH-average vs. $PM_{10}$-average (1), MLH-median vs. $PM_{10}$-median (2), MLH-average vs. $PM_{10}$-median (3), MLH-median vs. $PM_{10}$-average(4).

correlation options and MLH retrievals are with the exceptions of sites #32 (outskirts), #171 (urban background) and #174 (traffic) small: for a given MLH retrieval (i.e. same color) the range of $R$ (maximum minus minimum) for different options is typically 0.08, for a given option (i.e. same vertical line) it is 0.11 on average. For the three sites mentioned the sensitivity to the correlation option is however 0.25–0.35. The reason is that correlations involving averages of $PM_{10}$ (first and last vertical line of each block) clearly differ for those based on medians. The latter are less effected by short episodes of extreme concentrations which are not unusual for particulate matter.

As already mentioned these correlations are based on ceilometer measurements at one site, and that it was impossible in the framework of BAERLIN2014 to verify that the diurnal cycle of the MLH within the 20 km range of the air quality stations is identical. Large differences of the correlation coefficients are however also found if we restrict ourselves to the five BLUME-stations (#220, #143, #171, #174, and

#124) that are closest to the ceilometer site ($0.9 \leq d_{42} \leq 6.4$ km, see Table 1). Over this small area in the center of the city changes of the diurnal cycle of the MLH are very unlikely. Nevertheless our previous conclusions are confirmed: as can be seen from Fig. 9 the correlations between MLH and $PM_{10}$ are quite variable ranging from more than $R$=0.7 (#124, $d_{42}$=6.4 km) to less than $R = -0.7$ (#143, $d_{42}$=2.5 km).

At first glance it seems to be surprising that even within the same catagory the correlations are quite different. The three stations at the margin of Berlin (outskirts) show however different characteristics with respect to their distance to major traffic sources. Station #32 (Grunewald) is only 0.8 km west of the AVUS-motorway, whereas stations #77 and #85 are more than 3.5 km from the next motorway. The latter station is close to a large lake. Thus there is in principle sufficient time for mixing during the transport from these sources towards the measurement site, of course depending on the wind direction that certainly changes during to observation period. The three urban background stations show even more pronounced differences. For station #10 the distance to the next main road is larger than for the other two sites, and due to the east-west orientation and the broad street ventilation is more effective than for the reference site #42 (Neukölln) with a lot of vegetation in a typical residential neighborhood in the inner part of a big German city, and a comparably large distance to major roads. In contrast station #171 is close to a main road but it benefits from a good ventilation from the river Spree. For the traffic stations technical conditions, e.g., the number of lanes, the presence of traffic lights close to the monitoring station, and height and distance of the surrounding buildings becomes especially relevant because of the short distance between the emitters and the monitoring station. Consequently, the vertical dilution in the mixing layer is less relevant for $PM_{10}$ concentrations, and correlations are rather governed by the diurnal cycle of the traffic which is not necessarily dominated only by the morning and evening rush hours, but could have a significant contribution from buses and trucks throughout the day.

We conclude that the completely different correlations between mean diurnal cycles of MLH and $PM_{10}$ at the different sites as shown in Fig. 9 clearly demonstrate that the surface concentration of particulate matter is not determined by the vertical stratification of the mixing layer alone, but also by local sources and sinks and the wind field (see e.g. Tandon et al. (2010); Tai et al. (2010)). Moreover, the distance between the main sources and the measurement site is relevant.

The lack of a unique correlation is confirmed if we consider sub sets of data with specific meteorological conditions. Two examples are briefly discussed: the consideration of the wind field and the differences of working days and weekends. If only days are considered when the average wind speed over Berlin was below a certain threshold a pronounced correlation is more likely because the vertical exchange can dominate advection. Hourly wind measurements in 10 m altitude were available at three stations in Berlin, i.e. Tegel (52.5644° N, 13.3088° E; $d_{42} = 11.8$ km), Tempelhof (52.4675° N, 13.4021° E, $d_{42} = 3.1$ km) and Schönefeld (52.3807° N, 13.5306° E, $d_{42} = 13.9$ km). They constitute a

**Table 3.** Correlation coefficients $R$ between medians of hourly MLH (derived from COBOLT) and $PM_{10}$ for different sub sets of data: "all": diurnal cycles based on 67 days as shown in Fig. 9 (second vertical line of each block), "v40", "v30" and "v25": only consideration of days with average wind speed $\overline{v}$ below 4.0 m/s, 3.0 m/s, and 2.5 m/s, respectively, "m–f": Monday to Friday, "w-end": weekend only. The station IDs are according to Table 1.

| station ID | all | v40 | v30 | v25 | m–f | w-end |
|---|---|---|---|---|---|---|
| # 32 | 0.12 | 0.11 | -0.05 | -0.29 | 0.26 | -0.46 |
| # 77 | -0.35 | -0.41 | -0.40 | -0.44 | -0.12 | -0.80 |
| # 85 | -0.45 | -0.50 | -0.45 | -0.42 | -0.25 | -0.74 |
| # 10 | -0.71 | -0.77 | -0.84 | -0.83 | -0.68 | -0.76 |
| # 42 | 0.62 | 0.54 | 0.20 | -0.18 | 0.68 | -0.14 |
| #171 | -0.13 | -0.25 | -0.62 | -0.74 | 0.13 | -0.64 |
| #117 | 0.55 | 0.42 | 0.28 | -0.10 | 0.52 | -0.11 |
| #124 | 0.71 | 0.74 | 0.67 | 0.52 | 0.76 | 0.19 |
| #143 | -0.80 | -0.81 | -0.80 | -0.76 | -0.63 | -0.74 |
| #174 | 0.50 | 0.35 | -0.29 | -0.47 | 0.58 | -0.68 |
| #220 | -0.36 | -0.42 | -0.53 | -0.55 | -0.27 | -0.53 |

northwest to southeast transect through Berlin (see black stars in Fig. 1). For a simplified categorization of the wind field we use the daily averages of the wind speed $\overline{v}$. We found 52 (out of 67) days where $\overline{v}$ was below 4 m/s at all three stations, 28 days with $\overline{v} < 3$ m/s and only 16 days with $\overline{v} < 2.5$ m/s. In the latter case correlations between $PM_{10}$ and $H_{ml,v,L3}$ or $H_{ml,v,Q3}$, respectively, suffer from the low

number of successful retrievals (see Sect. 4.3). Therefore the correlation coefficients shown in Tab. 3 (columns "v40", "v30" and "v25", respectively) only refers to COBOLT-retrievals of the MLH. Though the correlation coefficients are in general more shifted to negative values compared to Fig. 9 (see also column labeled "all" in Table 3) and anti-correlations occur more frequently, the large spatial variability remains.

If we distinguish – as the second example – working days and weekends (columns "m–f" and "w-end" of Tab. 3, respectively), we find very pronounced differences with a tendency to stronger a anti-correlation for weekends. This is plausible as the diurnal cycle of the emissions is less pronounced. However, there were only ten weekends with ceilometer measurements during BAERLIN2014, so these findings should be treated as preliminary.

As an additional example one may focus on day/night differences of the correlation. For this purpose we use co-incident hourly measurements (depending on the ceilometer retrieval up to 1608 values) rather than the mean diurnal cycle as before to overcome the small number of samples. We define "day time" as the period between 07:01 CET and 20:00 CET, and "night time" as times before 07:00 CET

and after 21:00 CET. Then, for day time measurements we get very low correlation coefficients $-0.33 < R < 0.10$, for night time the correlation is only slightly different ($-0.27 < R < -0.09$). The main result is that during night time $R < 0$ for all sites, and only one site with $\|R\| < 0.1$ was found. On the one hand these values are plausible as we expect an anti-correlation between MLH and $PM_{10}$ in view of the suppressed vertical mixing in particular during night when the mixing layer is typically shallow (see Fig. 5). On the other hand the absolute values of $R$ are too small for supporting a strict scientific interpretation.

We conclude that the heterogeneity of the city is obviously more relevant than the selection of the MLH retrieval and the correlation option. The introduction of only three classes of monitoring stations (traffic, urban background, outskirts) cannot reflect the full complexity of pollution in the metropolitan area, and a re-assignment might be advisable when traffic flows have changed over years.

### 5.2 Correlation between MLH and gaseous pollutants

With respect to gaseous pollutants we restrict our discussion to $O_3$ and $NO_x$. Ozone measurements on a hourly basis are available at seven sites, $NO_x$ at all 16 sites (see Table 1).

The mean diurnal cycle of $O_3$ is shown in Fig. 10 for the five stations located at the outskirts of Berlin (dotted lines) and two urban background sites (#10 and #42, solid lines); medians considering 67 days of data are plotted. It exhibits the typical pronounced diurnal cycle with a maximum of about 100 $\mu$g/m$^3$ between 14:00 and 16:00 CET. The minimum occurs shortly after sunrise which was between 04:00 and 05:00 CET during the BAERLIN2014-campaign. Note, that the diurnal cycles based on averages instead of medians are quite similar: during the afternoon (largest concentrations) averages are about 5 $\mu$g/m$^3$ larger than medians. There is a close agreement between all stations, not only for the mean diurnal cycle but also on a daily basis (not shown here) suggesting that the spatial dependence of ozone concentration is less pronounced. This can be expected as ozone is not emitted but formed in the atmosphere within several hours after release of precursors. Thus, transport and mixing are key driving forces.

The diurnal cycles for $NO_x$ concentrations are shown in Fig. 11, again calculated as medians. The concentration at the stations at the outskirts of Berlin (dotted lines) are the lowest with a maximum during the morning rush hours of not more than 25 $\mu$g/m$^3$. The urban background stations (solid lines) show larger concentrations with a morning maximum of up to about 40 $\mu$g/m$^3$. Significantly higher concentrations are observed at the traffic stations (dashed lines), again with a maximum during the morning rush hours. The absolute values and the development during the day are however much more diverse than at the less polluted locations. One reason can be that roadside $NO_x$-concentrations depend strongly on the distance from the source (e.g. Bonn et al., 2016; Richmond-Bryant et al., 2017), a similar situation as for $PM_{10}$. Due to the spatial variability of the mean diurnal cycles it is clear that for the traffic sites the correlations must vary as well. If averages instead of medians are considered $NO_x$-

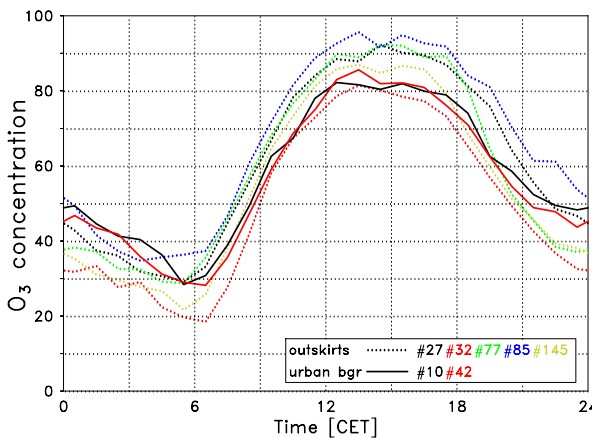

**Figure 10.** Diurnal cycle of $O_3$ concentration [in $\mu g/m^3$] at five stations at the outskirts (dotted lines) and two urban background stations (solid) as given in the legend (see also Table 1). Medians of the concentrations are plotted.

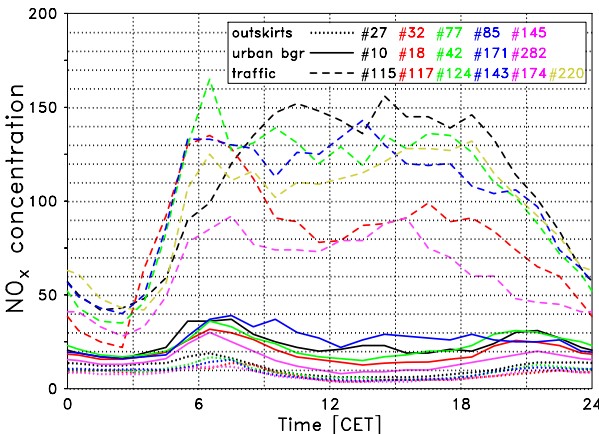

**Figure 11.** Diurnal cycle of $NO_x$ concentration [in $\mu g/m^3$] at all BLUME stations: five at the outskirts of Berlin (dotted lines), five urban background stations (solid) and six traffic stations (dashed) as indicated in the legend (see also Table 1). Medians of concentrations are plotted.

concentrations are somewhat larger (between 5 and 20 $\mu g/m^3$) and the morning maxima are slightly more pronounced.

Correlations between MLH and concentrations are shown in Fig. 12, separately for the three site-categories. For the outskirts of Berlin (leftmost block) and the urban background sites very strong pos-
5    itive correlations for ozone (circles) are derived. On average we find $R = 0.94$ for all sites and MLH retrievals. The differences between the sites are virtually negligible. One of the reasons for the very high

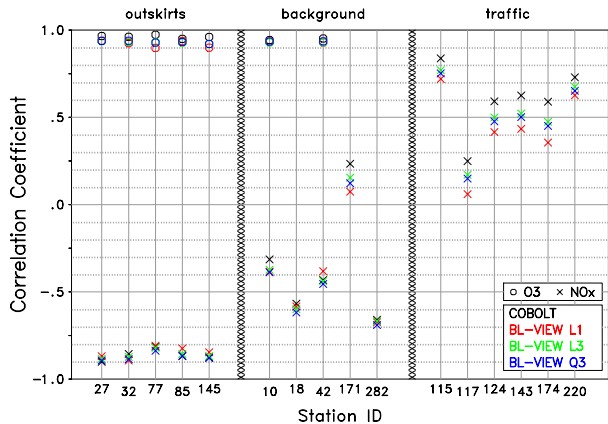

**Figure 12.** Correlation coefficient $R$ of mean diurnal cycles of MLH and $O_3$ (circles) and $NO_x$-concentrations (crosses), respectively, shown for the 16 sites as indicated by the ID-number according to Table 1. The four MLH retrievals are color-coded according to the legend. Correlations based on MLH-averages and $O_3$- and $NO_x$-medians are plotted. Note, that at the traffic stations no $O_3$-measurements are available.

correlations is that both MLH and $O_3$ concentration strongly increase after sunrise. The increase of the concentration is caused by the onset of photochemical ozone production and by downward mixing of ozone from the residual layer in the morning hours when the mixing layer grows because of radiative heating of the ground and increasing convection. As shown by Fallmann et al. (2016) and Kaser et al.

(2017) downward mixing of ozone from aloft can be a major source of near surface ozone for polluted urban sites with high $NO_x$ levels. In "green" areas of low $NO_x$ concentration ozone production is also intensified. So, high correlations can be found, though manifold and partly different physical reasons are responsible.

The correlations between MLH and $NO_x$ concentration at the sites on the outskirts are strongly neg-

ative, on average $R = -0.86$ is found. Again the spatial differences are almost negligible. An anti-correlation can be expected from mixing during the transport from the city center to the outskirts of Berlin. Negative correlations are also found for the urban background stations with the exception of station #171: due to their closer proximity to the main sources the absolute values on average are however slightly smaller ($R = -0.51$). Though labeled as "urban background" site #171 resembles much more

the traffic sites. As already mentioned in Section 5.1, it is indeed close to a major road, but in contrast to the $PM_{10}$-concentration the presence of the nearby river does not counteract in a similar way the $NO_x$-distribution. For sites dominated by traffic a positive correlation is found, but with a wide spread of values from $R \approx 0.16$ for site #117 to $R \approx 0.77$ for site #115. Additionally, there is a pronounced dependence on the MLH retrieval, e.g. for site #174 $R = 0.36$ (L1 retrieval) and $R = 0.59$ (COBOLT).

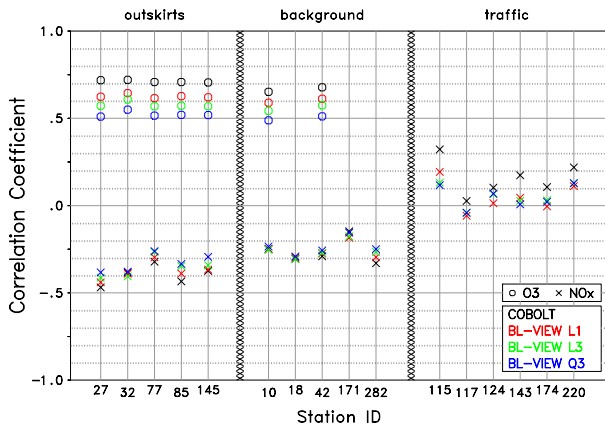

**Figure 13.** Same as Fig. 12 but correlation coefficient based on 1-hour measurements.

These findings are confirmed by investigations on an hourly basis (up to 1608 cases, see Fig. 13). Correlation between MLH and $O_3$-concentration (open circles) are again high and virtually independent on the location. However, differences between the MLH retrievals are in the order of 0.2: for BL-VIEW Q3 on average $R = 0.52$ with a very small variation with the location (standard deviation of 0.02) is found,

whereas the correlation is larger ($R = 0.71 \pm 0.01$) if COBOLT is applied. With respect to $NO_x$ concentrations the correlation coefficients are approximately $-0.36$, $-0.25$ for outskirts and urban background stations, respectively, and much more dependent on the site. For the traffic sites the correlation is weak with a large spread of $-0.1 \leq R \leq 0.3$. These results suggest that only in case of secondary compounds and primary pollutants in the absence of nearby traffic sources strong correlations between MLH and

gaseous pollutants can be found.

## 6  Summary and conclusions

The height of the mixing layer (MLH) is expected to have an influence on air quality at the surface. It is assumed that extended mixing layers lead to dilution of pollutants and thus tend to decrease surface concentrations. Several publications have indeed reported such anti-correlations. However, neither the

representativeness of such correlations for metropolitan areas nor the role of choice of the MLH retrieval have yet been investigated. This paper is devoted to these topics by examining the relationship between MLH and near surface concentrations of particulate matter ($PM_{10}$), $NO_x$ and $O_3$. It is based on two months of data from the field campaign BAERLIN2014 conducted in Berlin, Germany.

Frequently used tools to determine the MLH are automated lidars and ceilometers (ALC). Espe-

cially commercial systems with their unattended continuous operation are very promising since they

are available as networks. Here, we compare four different approaches to determine the MLH, three of them based on proprietary software delivered by the manufacturer of the instrument (Vaisala), and the recently developed approach COBOLT (Geiß, 2016). It was found that a complete diurnal cycle with a high temporal resolution often cannot be derived from the proprietary software, and that there is a tendency to overestimate the MLH in the presence of the residual layer.

It is obvious that the differences of the retrieved MLH influence the correlation coefficients between MLH and pollutant concentrations. For mean diurnal cycles correlation coefficients differ by approximately 0.1 if different MLH retrievals are applied. These differences are smaller than the differences found when different locations in the city are compared – even if their distance is only a few kilometers from each other. In case of $PM_{10}$ we found strong correlations as well as strong anti-correlations even if the sites are assigned to the same category (e.g. "urban background" or "traffic stations"). This clearly demonstrates that the MLH is not the only parameter controlling the surface concentration, and that local emissions and transport play a dominant role. This is in agreement to the pronounced heterogeneity over Berlin as reported by Bonn et al. (2016). In case of ozone as a secondary pollutant the correlations for different sites show only small differences. The strong correlation was found due to the similarity (although for different reasons) of the mean diurnal cycles of ozone and MLH with maximum values in the afternoon. An anti-correlation for near-surface concentrations of $NO_x$, as suggested by several previous studies, was only found in the absence of direct exposure to traffic sources.

We conclude that in case of a large city as Berlin the MLH can be an indicator for urban air quality only in a very limited sense, and that any correlation between MLH and concentrations of pollutants should be treated with care: it is unlikely that they are representative for the entire metropolitan area, in particular if the terrain is flat. At least for the observed summer period in Berlin this was not the case. Consequently, whenever links between MLH and near-surface concentrations are interpreted, it is mandatory to carefully describe the location, i.e., meteorological conditions, local sources etc., and the details of the MLH retrieval. Compared to the heterogeneity of the former we think that the selection of a certain MLH retrieval does not have the highest priority for correlation studies. It would be interesting to study winter time conditions when the $PM_{10}$ concentrations in Berlin are about 50 % higher than in summer. We do not expect that in winter the MLH is the only controlling parameter, but it is not clear if the correlation (and its variability) is more or less pronounced. It remains open whether the situation is different for regions without pronounced changes in land use, without significant local emissions, or in areas with pronounced orography.

To better understand the complex interactions between the MLH, wind field, emissions, chemical processing etc. for air quality, there is a need for models down to a building-resolving scale as well as more extended data sets especially for heterogeneous areas. The specific setup of models and experiments must be defined according to the scale of interest. Continuous ceilometer measurements including at

least one complete annual cycle can provide a significant contribution and help to investigate the generality of the results, e.g. to check for seasonal changes or for differences between working days and weekends. It is obvious that ALC with a very low overlap range are required for the observation of very shallow mixing layers typical during night time and in winter. Moreover, it would be nice to have
co-incident ceilometer measurements at different sites or to have one or more mobile systems to check our hypothesis that the MLH does not change on a scale of a large city.

It should be added that accurate retrievals of the MLH are beneficial for several applications: they can be used for box-model calculations and for the validation of meteorological models and the meteorological part of chemistry transport models. As the MLH is not a prognostic variable it is important
to assess the accuracy of different parameterizations (e.g. Hu et al., 2010; Gan et al., 2011; Svensson et al., 2011; Banks et al., 2016). In this context a high accuracy of the MLH retrieval is crucial and a methodology that provides the full diurnal cycle with high temporal resolution, and avoids wrong allocations of aerosol layers must be applied. Finally, we want to emphasize that state-of-the-art ALC allow for the derivation of profiles of the particle backscatter coefficient $\beta_p$ if the signals have been
calibrated (e.g. O'Connor et al., 2004; Wiegner and Geiß, 2012). In case of ceilometers emitting in the spectral range near 910 nm, the signal must however be corrected for water vapor absorption (Wiegner and Gasteiger, 2015). Profiles of $\beta_p$ can be used for the validation of chemistry transport models (e.g. Emeis et al., 2011) in a more direct way than the MLH as e.g. mixing ratios or mass concentrations of aerosol particles (or different aerosol components) are available as prognostic variables. Applying the
adequate scattering theory, $\beta_p$ can then be derived. On the long term perspective this is the preferable strategy for validation.

*Acknowledgements.* We like to thank Carsten Jahn (KIT/IMK-IFU) and Moritz Pickl during his internship at KIT/IMK-IFU for careful instrument operation and data analyses. We are grateful to Albrecht von Stülpnagel and Martin Schacht (both at Senate Department for Urban Development and the Environment, Berlin) for data-provision
and technical assistance, and Holger Gerwig (German Environment Agency, Dessau) for helpful discussions.

The wind measurements discussed in Section 5.1 were obtained from the Climate Data Center of German Weather Service (DWD).

Alexander Geiß' contribution to this paper has received funding from the European Union Seventh Framework Programm (FP7/2007-2013) under grant agreement No. 262254. Ka Lok Chan was supported by the Marie Curie
Initial Training Network of the European Seventh Framework Programme (Grant No. 607905).

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
