# Peer review of "Mixing layer height as an indicator for urban air quality?"

_Atmospheric Measurement Techniques, 2017_

## Referee Comment (RC1) · Anonymous Referee #1 · 13 Apr 2017

General comments:

In this generally well-written paper, ceilometer-based mixing heights are derived, using three versions of the proprietary software BL-VIEW provided by the manufacturer, Vaisala, and an own algorithm, COBOLT. The latter shows fewer cases where the residual layer is mis-interpreted as the mixing height. Furthermore, the relationship between MLH and near surface pollutant concentrations has been investigated. Whereas for ozone a clear correlation was found, PM10 and NOx show more complex patterns, indicating that the mixing height is not the only parameter influencing the levels of these pollutants.

While these results are plausible and well explained, the paper currently suffers from a few serious deficiencies. First, the investigation comprises only two summer months,

and day-night differences in the relationship MLH and pollutants are not at all considered. Second, the whole area of Berlin is represented by only one ceilometer. It is however well known that the mixing height will show some variation over such a large area both day and night, depending on the degree of urbanization and other surface-related influences. A discussion on this issue is needed. The short investigation period and the use of only one ceilometer are currently briefly discussed in the conclusions, promising to tackle these issues in the future. However, these shortcomings have to be discussed more deeply, including references. An investigation of day-night differences has to be included in a revised version. Some of the figures need improvement (see below).

Specific comments:

p. 3, line 1: The statement is too optimistic (frequently used approach); if true, provide more references. I think determining MLH from ceilometers in a reliable manner is quite a new subject.

p. 3, line 20: The short investigation period is mentioned here for the first time (see General commentsp. 4, line 7: The main shortcoming of Sodar and RASS is that they usually cannot provide the whole diurnal cycle of MLH in Central Europe, especially in summer. A Sodar alone can give a reliable estimate of MLH only with careful data analysis, see e.g. Bound.-Layer Meterorol. 124, 3-24 (2007).

p. 4, lines 17-18: The advantage of spatial coverage of a network of ceilometers is not used in this study.

p. 8, line 5: please elaborate statement (one ceilometer is representative for a metropolitan area; see also General comments).

p. 10, last paragraph: A graphical sketch (Fig. 2 is not sufficient) on how the COBOLT algorithm works would facilitate understanding. How is "the parameter" defined?

p. 12, discussion of Fig. 2 (bottom): from visual inspection, L1 seems to work best in

comparison with COBOLT. Do the quality flags really improve the comparison? This aspect is not discussed.

p. 18, top: Only one ceilometer: this is indeed the main drawback of the investigation (see also General comment).

p. 24, lines 11 and 12: Robustness and representativeness are also not really investigated in this paper.

Technical corrections:

p. 1, line 5: . . .has been investigated p. 1, line 7: July and August p. 2, line 9: . . . and when meteorological conditions . . . p. 2, line 13: mass concentrations p. 2, line 27: either "for a chemical box model" or "for chemical box models" p. 3, line 3: In particular, p. 3, line 4: are established, p. 5, line 8: These findings p. 19, Fig. 7: The lines for the outskirts stations are missing p. 21, line 13: probably: . . . larger at the outskirts sites. p. 22, Fig. 9, Fig. 10: The lines for the outskirts stations are missing

---

## Referee Comment (RC2) · Anonymous Referee #2 · 25 Apr 2017

General comments

The paper 'Mixing layer height as an indicator for urban air quality?' investigates the relationship between ceilometer retrieved mixing layer heights and near surface pollutant concentration. Three different versions of BL-VIEW and a novel approach called COBOLT were used to derive the Mixing Layer Height at one selected station in the urban area of Berlin. Whereas for ozone the approach seems to show meaningful results, NOx and PM10 reveal a much more complicated picture and MLH might be difficult to be defined as main factor influencing near surface levels. Overall, the paper is well written and easy to follow, but however needs some more critical discussion on certain points.

In my point of view, using just one ceilometers/ location might not be sufficient to answer

the question given in the title. It is clear, that it is difficult to extent the study to other locations at that stage, but however, this aspect should be discussed more in detail. As highlighted by reviewer 1, I share the opinion that a day/night comparison might be interesting.

For meteorological conditions being a main driver of turbulent mixing it might be interesting to include some meteorological observations characterizing the measurement location and selected study period. With a observation height of about 5m it might be interesting, which amount of the measured concentration is originated from the actual location and which amount is advected from neighboring areas or 'removed' by vertical mixing. Here again a night/day difference would be interesting. How does this aspect influence on the analysis at one selected point?

Specific comments:

p.2 line 3-4: Is there evidence in your study? Otherwise put this sentence in the introduction or conclusion.

p.2 line 19: measurements, data instead of techniques

p.2, line 27: box models

p.3, line 16: COBOLT: add one sentence highlighting novelty, functionality

p.3, line 19: aim to instead of may

p.4, line 1: specify 'active remote sensing networks' (e.g....)

p.5, line 15: chemical processes? What about Ozone? Where does it come from – downward mixing, secondary formation?

p.5 line 28-31: This is not part of your analysis and could be moved to the conclusion.

p.6, line 9: specify 'secondary material'

p.6, line 14: hourly measurement

p.8, line 1: how representative is the measurement location in 5m height for near sur­face pm10 concentration? How does this impact the representativeness for the MHL measurements for this area?

p.16 Figure 5: legend has to be added

p.17 line 2 ff: this chapter defines the scope of the study and in my opinion appears to late in the manuscript which results in a misbalance between introduction/methods and results. The first part until 5.1is more an introduction to a new topic than a presentation of results. I might be helpful to include some of these aspects ion the introduction (without changing the whole manuscript). Line 2: Ozone and NOx also measured at BLUME?

p.18 line 8: it is unclear on which basis the median was calculated. 67 measurements each hour at every station? p.18, line 16-20: can you proof your assumptions by adding meteorological observation here? Is there a secondary circulation generated by the Urban Heat Island? Please specify the term 'meteorological interpretation'.

p.20, line 31f: see comment above

p.21 general: here you mention briefly the problem of point measurements. This aspect could be further discussed. It is interesting if there is a mismatch between the timing of MLH and air quality observation. Does a low MLH mean a high concentration at the same time? What is the order of the processes? Where do meteorological conditions come into play?

Chapter 6 It is not the extended mixing layer itself which is the initial precursor of dilution of pollutants near the ground. Several processes interact which each other which might as well lead to an extension of the mixing layer height.

---

## Referee Comment (RC3) · Anonymous Referee #3 · 5 May 2017

This is an interesting manuscript as it discusses the relationship of the mixing layer height (MLH) and near surface pollutant concentrations. The authors perform correlations of the MLH and PM10, NOx, and O3, and found varying results. The authors believe that the effects of the heterogeneity of the emission sources, chemical processing and mixing during transport exceed the differences due to different MLH retrievals.

With regard to the use of the different MLH retrieval methods (Vaisala proprietary software, COBOLT), which are solely based on aerosol backscatter signal, I was wondering, if radiosondes have been used for a conclusive validation during the BAERLIN campaign. Also, I was wondering why other methods such as the Haar wavelet method or a cluster method have not been considered/discussed.

With regard to the relation of the MLH with air quality, it is well known that the local

change of any given pollutant is not only controlled by the MLH, but by a combination of emission, chemical transformation, removal, advection, convection and turbulent mixing. Also, it is known that at the microscale level urban structures cause flow disturbations and thus deviations from the mean air quality of a larger, representative fetch in an urban area. An example is the well-known wind rotor system in street canyons. Thus the relationship of the MLH with surface concentration critically depends on the fetch area representative for a given measurement site. These well-known processes are not properly addressed in the paper. Due to the rather flat larger area of Berlin, it can be expected that transport processes may play a dominant role in the distribution of pollutants, both at the mesoscale and microscale level. I am surprised to see that the authors did not consider any of the findings associated with the BERLIOZ experiment in 1998 (mostly published in Journal of Atmospheric Chemistry 42, 2002, but also others), which focused on the upwind-downwind conditions found for the Berlin case, as well as the pollutant concentrations within the boundary layer and aloft in the same area and the impact of long-range transport. I would not expect an unambiguous relationship between the MLH and surface concentrations at any given location and under any given meteorological situation in the Berlin area. Rather, I would only expect a dominant role of the MLH on surface concentration, when advection is at a minimum, i.e. under stagnant wind conditions. In its current form this paper neglects the discussion of the MLH with regard to different wind regimes, both with regard to wind speed as wind direction. It should also be mentioned that not only pollutants can be transported, but also physical properties of the boundary layer including the MLH depending on the history of air masses. This extended in-depth analysis is a crucial requirement for a potential publication in AMT.

This is a list of more specific, mostly minor issues, which need to be addressed:

Page 2, L25-28: The paper by Czader et al. (2013) should be added as it is one of the earlier examples to use ceilometer derived MLHs for validation in conjunction with comprehensive air quality modeling.

Czader et al. (2013): CMAQ modeling and analysis of radicals, radical precursors and chemical transformations, J. Geophys. Res., 118, 11,376-11,387, doi: 10.1002/jgrd.50807

Page 5, L18-19: I think both terms MLH and Hml mean the same. I suggest to use one term throughout the entire text.

Page 5, L22: Please define what "width" would mean exactly: horizontal or vertical?

Page 6, L9: Please define what is meant exactly by "secondary material"?

Page 7, L1-3: "These data....in whole Germany". Is this statement important in understanding the contents of the paper? I suggest to remove it.

Page 8, L4: What "information" is exactly meant?

Page 8, L6: Suggest to remove ", which is one hour different to UTC.", as UTC is not being used in the paper.

Page 10, L16: Please explain what is actually meant by "cross-platform" here, and why it would be helpful?

Page 15, L26: "Concentration measurements" of what?

Page 15, L30: "The latter...(Pappalardo et al., 2014)". Please explain the schedule of EARLINET and explain whether the BAERLIN approach was important for the EARLINET approach or the other way round (which is more likely).

Page 15, L32-33: Please mention that these specific COBOLT results refer to the entire campaign period.

Page 17, L11-12: What is exactly meant by "All measurements are performed under ambient conditions"? They way it is written it would mean that the air quality station was not air-conditioned.

Page 17, L18: I think this "significant horizontal heterogeneity" refers to surface mea-

surements here. Please clarify.

Page 17, L15-16: What is exactly meant by "inorganic species": gas-phase, particle bound or both?

Page 17, L17: The reference "von Schneidemesser et al., 2017" is still in preparation and therefore not citable.

Page 17, L17-18: "Here we do not discuss these topics...". In this case please remove the preceding L13-17 as they are not within the scope of this paper.

Page 18, L26-28: What is the justification for using these different correlations? The statistically most reliable quantity would be the median anyway, as it minimizes the impact of outliers. This is in particular true for such a quantity as PM10, which is mostly primarily emitted.

Page 20, L23: "...with a lot of vegetation, a high density of buildings...". This sounds like a contradiction: where there is high density of buildings how can there be lot of vegetation at the same time?

Page 20, L17-18: The authors mention aerosol formation. Would PM10 data provide any indication for aerosol formation? If so, please explain.

Page 20, L17: The authors mention that relative humidity may have an impact on PM10. Would PM10 concentration decrease or increase with relative humidity?

Page 21, L1-2: What classes in addition would the authors recommend?

Page 21, L2-4: This statement is obvious and has been considered in many urban air quality networks over many decades.

Page 21, L31 - Page 23, L5: It is well-known that O3 can be mixed from the residual layer into the convective layer, also for the case of Berlin (e.g. see BERLIOZ special issue in the Journal of Atmospheric Chemistry 42, 2002). The excellent correlation of the MLH with ozone in urban areas may not be surprising at all, as both processes are

ultimately driven by incoming solar radiation provided there are sufficient precursors for O3 formation available. In other words the relation between the MLH and O3 is apparent, but not causally determined. This should be mentioned.

Page 24, L7: "Whether ...studied". I suggest to remove this sentence. It is obvious that the potential impact of the MLH on ambient concentrations decreases with decreasing distance to the corresponding emission source.

Page 24, L13: As I remember Xu et al. (2011) do not report MLH observations and thus no correlation with primary or secondary pollutants.

Page 25, L10-12: I guess it is well-known that there is not one only parameter which controls surface concentrations.

Page 25, L27-28: In their paper the authors have tried to argue that MLH is not the only parameter which controls surface pollutant concentrations. Why then would it be of interest to perform a winter study in Berlin and why is it of importance that PM10 concentrations are 50% higher in winter compared to summer in Berlin. If there is no consistent correlation of MLH with PM10 in summertime why should it be different in wintertime?

Page 26, L4-6: The authors state that MLH data is beneficial for box-model calculations and validation of chemistry transport models. While I would agree on the authors' statement in this sentence I do not completely understand what the authors' justification would be for this, since according to their paper the authors largely argue that there is no consistent correlation of the MLH with air pollutants. This should be clarified.
* * *

---

## Author Comment (AC1) · 22 Jun 2017

**Authors' response to reviewer #1**

We thank reviewer # 1 for carefully reading our manuscript and the provision of many useful comments and detailed suggestions. We have considered all comments. They gave as useful hints where improvements of the paper were necessary to better understand our methodology and conclusions. Below all points raised by the reviewer are repeated; our comments are added in italics.

The changes (revised version vs. AMTD-paper) are highlighted as displayed by latexdiff ("diff.pdf", maybe renamed when uploaded as a supplement). For the sake of clarity only small changes are explicitly mentioned in our point by point replies, otherwise we refer to the corresponding parts of "diff.pdf" (in blue). Note, that some of our responses interact with comments of the other reviewers, so sometimes it is difficult to refer a change to one specific reviewer's comment.

**Point by point replies**

**General comment**

[...]

While these results are plausible and well explained, the paper currently suffers from a few serious deficiencies. First, the investigation comprises only two summer months, and day-night differences in the relationship MLH and pollutants are not at all considered. Second, the whole area of Berlin is represented by only one ceilometer. It is however well known that the mixing height will show some variation over such a large area both day and night, depending on the degree of urbanization and other surface related influences. A discussion on this issue is needed. The short investigation period and the use of only one ceilometer are currently briefly discussed in the conclusions, promising to tackle these issues in the future. However, these shortcomings have to be discussed more deeply, including references. An investigation of day-night differences has to be included in a revised version. Some of the figures need improvement (see below).

> → *The main concern of the reviewer is the limited length of the observation period, the number of ceilometers and the missing discussion of day-night differences (a similar comment was given by reviewer #2).*

*The major focus of the BAERLIN2014 project was on ozone, secondary organic aerosol and the effect of urban vegetation. All of these effects are found at its maximum at summer especially at highest temperatures and oxidation strength. Because of the limited amount of resources the campaign must be concentrated on three months (June, July and August; this remark as a background information).*

*In principle we agree with the reviewer that more ceilometers would have been beneficial for the study. As already mentioned research projects are always limited with respect to money, personnel and hardware (temporal extent and spatial coverage of measurements, number of measured atmospheric variables, number of instruments, etc.). In case of BAERLIN2014 e.g. no external funding was available. As a consequence field campaigns always are limited in time: this was also true for the BERLIOZ campaign mentioned by reviewer #3.*

*Nevertheless we believe that BAERLIN2014 provided very valuable scientific results even if there was only one ceilometer available. We were able to demonstrate to what extent differences in MLH-retrievals play a role for calculating correlations between MLH and air quality parameters. By addressing standard retrievals (the proprietary software of the ceilometer manufacturer) and air quality measurements from an official monitoring network we think that the conclusions are relevant. These investigations could only be performed in the framework of a dedicated campaign because ceilometers do not yet belong to the standard equipment of urban air quality networks. To our knowledge only in Paris a network including three ceilometers for routine observations was recently established: the collaborative measurement platform "OCAPI". Results are not yet published. See extension of the introduction (page 3, lines 24 ff of diff.pdf):*

*Prospectively also the implementation of urban networks for air quality studies is likely at least for selected cities occasionally suffering from pollution events – recently three ceilometers were set up in larger Paris for this purpose (OCAPI: Observation de la Composition Atmosphérique Parisienne de l'IPSL).*

*Based on our research, open questions could be identified, one of them being the need for an in-depth investigation of the behavior of the mixing layer over a large municipality. So we hope that in future the wishes of the reviewer (and ours) to have more ceilometers and at least one full annual cycle of the MLH can be fulfilled, and that our paper will be*

*a motivation for setting up the corresponding infrastructure (see also our replies to the detailed comments of reviewer #1 below). In the conclusions (center of page 31 of diff.pdf) we have also stressed that numerical models (mesoscale, microscale) are required as well.*

*As a consequence we have added several sentences, in particular we have clearly describe the motivation of our study to avoid misunderstandings (see introduction, page 3, line 6 ff of diff.pdf).*

*Following the suggestions of reviewers #1 and #2 we have extended Sect. 5.1 by discussing day-night differences and the influence of the wind field (reviewer #3). In addition a short comment on differences between working days and weekends has been added (see pages 24–26 of diff.pdf).*

*Reviewer #1 was primarily interested in day-night differences: The resulting correlation coefficients $R$ of hourly values of MLH and $PM_{10}$ for all sites with $PM_{10}$ measurements are shown in Figs. 1 and 2 (next page). Figure 1 covers the time period between 07:01 CET and 20:00 CET ("day time"), whereas Fig. 2 is for measurements before 07:00 CET and after 21:00 CET ("night time"). The four MLH retrievals are color-coded according to the legend. It can be seen that the absolute values of $R$ are small as already stated in the original manuscript (for measurements of the whole day we have a range $-0.3 < R < 0.1$). For day time measurements (Fig. 1) we get $-0.33 < R < 0.10$ , for night time the correlation is slightly different ($-0.27 < R < -0.09$). The main difference is that during day time there are three out of 11 stations with positive correlations and 8 sites with $\|R\| < 0.1$, whereas during night time $R < 0$ for all sites and only one site with $\|R\| < 0.1$. These values are plausible as under ideal conditions an anti-correlation between MLH and $PM_{10}$ is expected in view of the suppressed vertical mixing when the mixing layer is very shallow during night. Note, that during night the number of point source decreases, in particular at the outskirts, so that we find the lowest absolute values there. Nevertheless the small absolute values of $R$ suggest that the MLH is not the only influencing factor. This is in accordance with several comments of all reviewers (and several statements in our manuscript) that there is no "simple" link between $PM_{10}$ and MLH, on the contrary many processes are relevant.*

*To avoid a substantial increase of the number of figures in the paper we have summarized the results of the above mentioned issues as a table*

[Figure]

Figure 1: Correlation coefficient $R$ of hourly values of MLH and $PM_{10}$ shown for the 11 sites as indicated (station ID according to Table 1). The four MLH retrievals are color-coded according to the legend. Only measurements between 07:01 CET and 20:00 CET are considered.

*and a new paragraph (see page 26 of diff.pdf and the new Table 3).*

[Figure]

Figure 2: Same as Fig. 1, but only measurements before 07:00 CET and after 21:00 CET are considered. NOTE: THESE FIGURES ARE FOR DEMONSTRATION ONLY, NOT INCLUDED IN THE MANUSCRIPT.

**Specific comments**

- p. 3, line 1: The statement is too optimistic (frequently used approach); if true, provide more references. I think determining MLH from ceilometers in a reliable manner is quite a new subject.

    → *We have added more references: Haman et al. (2012), Caicedo et al. (2017) (see also response to reviewer # 3), de Bruine et al. (2017). A paper by Knepp et al. has been submitted to AMTD on 30. May 2017, and cannot be cited yet. More citations can be found in the already cited papers. Moreover we have added a short remark on "ceilometers" and "ALC" (automated lidars and ceilometers), see introduction, page 3 of diff.pdf.*

    *In the last years several hundreds of ceilometers and ALC have been set up – not only by weather services but also by universities and research institutes. This triggered several activities to develop MLH-retrievals and improve their reliability – so it can be considered as a rather new subject. Most of the users rely on "atmospheric products" (primarily cloud bottom heights and mixing layer heights) that are automatically provided by the proprietary software, even if it is sort of a black box. Thus, from our point of view it makes sense to discuss associated problems, e.g., the risk to over-interpret the ceilometers' output (see our extended introduction, page 3 of diff.pdf). On the long term perspective it is likely that ceilometers will be the standard instrument to automatically monitor the aerosol distribution (this was the motivation of most weather services to establish these networks). We are sure that these instruments have a high potential if data are correctly exploited, as a consequence this subject will become even more relevant. Some future applications are briefly mentioned in the conclusions.*

- p. 3, line 20: The short investigation period is mentioned here for the first time (see General comments).

    → *The duration of the campaign is mentioned in the abstract, and in the introduction after a general discussion of the topic (p. 3, line 20). We felt that this was a logical structure, but it can also be moved to the end of page 2 (see page 3, first lines of diff.pdf).*

- p. 4, line 7: The main shortcoming of sodar and RASS is that they usually cannot provide the whole diurnal cycle of MLH in Central Europe, especially in summer. A sodar alone can give a reliable estimate of MLH only with careful data analysis, see e.g. Bound.-Layer Meterorol. 124, 3-24 (2007).

  → *We agree and added a corresponding comment to the manuscript including the suggested reference (Piringer et al, 2007, page 4 line 27 of diff.pdf). In Seibert et al. (2000) and Emeis et al. (2012) – already cited – this topic is also discussed.*

- p. 4, lines 17-18: The advantage of spatial coverage of a network of ceilometers is not used in this study.

  → *We regret that our statement could be misunderstood. We mentioned "networks" here to highlight that there is an infrastructure of active remote sensing instruments that is very dense compared to research lidar networks (e.g. compared to the spatial and temporal coverage of EARLINET). Urban ceilometer networks are not known to us – at the EGU 2017 one of the authors (MW) learned that recently a few new instruments have been set up so that small scale investigations might be possible in future for selected cities, provided that it is known to potential users. Currently the implementation (mainly for scientific or educational purposes at universities) is not coordinated. The only exception seems to be Paris (OCAPI). We have added a corresponding note (see our reply to the general comment).*

- p. 8, line 5: please elaborate statement (one ceilometer is representative for a metropolitan area; see also General comments).

  → *We have written "is assumed to be representative", not "is representative". With only one ceilometer (see also our other replies) we cannot prove that such a strong statement is true for Berlin. So we refer to a previous case study in Munich (see the citation), and investigations of the diurnal cycles of the MLH in Munich, Freising and Augsburg, which are almost identical (Geiß, 2016). The distance between these stations however is somewhat larger than the size of Berlin (approximately 50 km). Similar findings were published by Lotteraner and Piringer (2016): they compared*

*mean diurnal cycles for Vienna and Obersiebenbrunn, a village 26 km east of Vienna.*

*Moreover we know of a short case study in the greater area of Paris including a lidar site in the city center (Jussieu), one site with a lidar and a ceilometer at SIRTA (outskirts) and one lidar site 105 km south of Paris, supplemented by mobile lidar measurements from a van. This study shows similar diurnal cycles for Jussieu and SIRTA, and slightly lower MLHs at the rural site (see citation on page 9, line 6 of diff.pdf). These results support our assumptions.*

*Another argument for our assumption is that the terrain around Berlin is quite flat. As the situation might be different in areas with pronounced orography (e.g. a city in a valley) we have explicitly mentioned this here, in the conclusions (page 31, line 9 of diff.pdf) and in the abstract (page 2 line 2 of diff.pdf, see also reply to reviewer #2).*

*As the most important argument we want to stress that our conclusions also hold if only air quality measurements very close to the ceilometer site are considered (see detailed comment to "p. 18" below).*

- p. 10, last paragraph: A graphical sketch (Fig. 2 is not sufficient) on how the COBOLT algorithm works would facilitate understanding. How is "the parameter" defined?

  → *We have significantly extended the description of COBOLT including the most relevant equations. We believe that this extension is sufficiently clear to understand how COBOLT works, so that it is not necessary to add a flow chart to the manuscript as well. Examples of applications under different meteorological conditions and comparisons to independent data sets (e.g. radio sondes; see also comment of reviewer #3) would "overload" this paper and can only be presented in a separate publication. For modifications see pages 11-14 of diff.pdf.*

- p. 12, discussion of Fig. 2 (bottom): from visual inspection, L1 seems to work best in comparison with COBOLT. Do the quality flags really improve the comparison? This aspect is not discussed.

  → *Visual inspection cannot fully reveal all differences of the retrievals, especially as only one day is displayed in Fig. 2. This*

*figure is only shown to illustrate the problem and the different solutions. For quantitative conclusions we have included Fig. 3: Here, the differences for the whole period are plotted, separately for the different retrievals. It can be seen that consideration of the quality flag in particular reduces the number of cases where the retrieved MLH is (much) larger than the COBOLT-retrieval (panels b and c).*

*On the other hand, it is obvious that the number of successful retrievals is drastically reduced if the quality flags are used as described. With stricter requirements (e.g. quality flag must be 3, i.e. highest level) the number of MLH-retrievals drops from 8346 (if the quality flag is ignored) to 3331 (if only the highest quality is considered) to 2998 (if only the lowest candidate level is considered if it has the highest quality). This fact is described in Section 4.3 and is shown in detail (as a function of the time of the day) in Fig. 4c.*

- p. 18, top: Only one ceilometer: this is indeed the main drawback of the investigation (see also General comment).

  → *As already mentioned, we would have been lucky if more ceilometers had been available. This was however not the case and cannot be changed afterwards. On the other hand five air quality stations are very close to the ceilometer site (within 6.4 km, see Table 1). On this spatial scale changes in the diurnal cycle of the MLH are very unlikely, especially as all are in the center of Berlin and no environments like forests or lakes are included. If we restrict ourselves to only these sites (#220, #143, #171, #174, and #124) our conclusions remain valid: it is demonstrated in Fig. 8 that correlations between MLH and $PM_{10}$ are quite variable so that no generally applicable correlation coefficient can be found. We add a corresponding paragraph to the manuscript to emphasize this (pages 23 bottom to page 24 top of diff.pdf).*

  *With respect to the correlation between MLH and $NO_x$ the closest stations show positive R due to the strong contribution of traffic emissions. Ozone measurements were not available in the vicinity of the ceilometer site. We cannot exclude that at the outskirts of Berlin the MLH is different due to the different surface properties. The above mentioned investigations (Munich, Paris, Vienna; see reply to comment "p. 8 line 5") suggest that – if differences oc-*

*cur – they more likely concern the height of the mixing layer than the temporal development. So it can be assumed that the correlations are not very much affected. Nonetheless, for an ultimate clarification more ceilometers and further investigations would be highly desirable as mentioned in our conclusions.*

- p. 24, lines 11 and 12: Robustness and representativeness are also not really investigated in this paper.

  → *Our statement was related to previous studies and their shortcomings. But we agree with the reviewer that we have not covered all open questions in our paper: we have only shown for a limited period at one place that the correlations are not representative. And we have investigated the role of the MLH retrieval. So we dropped the word "robustness" to avoid misunderstandings (see* page 29 of diff.pdf, beginning of Sect. 6*).*

Technical corrections:

- p. 1, line 5: "...has been investigated"

  → *Corrected*

- p. 1, line 7: July and August

  → *The campaign started in June, however, the ceilometer was installed only by the end of June. To be consistent with other papers we would like to leave this sentence unchanged.*

- p. 2, line 9: "...and when meteorological conditions..."

  → *Changed*

- p. 2, line 13: mass concentrations

  → *Changed*

- p. 2, line 27: either "for a chemical box model" or "for chemical box models"

  → *Changed (second option)*

- p. 3, line 3: In particular,

  → *Changed*

- p. 3, line 4: are established,

  → *Changed*

- p. 5, line 8: These findings

  → *Corrected*

- p. 19, Fig. 7: The lines for the outskirts stations are missing

  → *We don't understand this comment: dotted lines are included. Maybe it is a matter of the resolution; however we have checked this on a printed page and it was readable. In case there are problems this can be fixed during the type-setting process.*

- p. 21, line 13: probably ...larger at the outskirts sites.

  → *What is meant is the following: if the diurnal cycle of $O_3$ concentrations based either on averages or medians are compared, the maximum values during the afternoon are higher by approximately 5 $\mu g$ $m^{-3}$ in case of averages (page 26, line 29 of diff.pdf, Sect. 5.2). This is valid for all sites where ozone measurements were available. That ozone concentrations at the outskirts (dotted lines) are in general higher (i.e. the comment of the reviewer) is also true.*

- p. 22, Fig. 9, Fig. 10: The lines for the outskirts stations are missing

  → *See reply above on Fig.7.*

Additional references:

- de Bruine, M., Apituley, A., Donovan, D. P., Klein Baltink, H., and de Haij, M. J.: Pathfinder: applying graph theory to consistent tracking of daytime mixed layer height with backscatter lidar, Atmos. Meas. Tech., 10, 1893-1909, doi:10.5194/amt-10-1893-2017, 2017.

- Caicedo, V., Rappenglück, B., Lefer, B., Morris, G., Toledo, D., and Delgado, R.: Comparison of aerosol lidar retrieval methods for boundary layer height detection using ceilometer aerosol backscatter data, Atmos. Meas. Tech., 10, 1609-1622, doi:10.5194/amt-10-1609-2017, 2017.

- Haman, C. L., Lefer, B., and Morris, G. A.: Seasonal Variability in the Diurnal Evolution of the Boundary Layer in a Near-Coastal Urban Environment, J. Atmos. Oceanic Technol., 29, 697710, 2012.

- Lotteraner, C. and Piringer, M.: Mixing-Height Time Series from Operational Ceilometer Aerosol-Layer Heights, Boundary-Layer Meteorol., DOI 10.1007/s10546-016-0169-2, 2016.

- Piringer, M., Joffre, S., Baklanov, A., Christen, A., Deserti, M., De Ridder, K., Emeis, S., Mestayer, P., Tombrou, M., Middleton, D., Baumann-Stanzer, K., Dandou, A., Karppinen, A., and Burzynski, J.: The surface energy balance and the mixing height in urban areasactivities and recommendations of COST-Action 715, Boundary-Layer Meteorol., 124, 3–24, doi:10.1007/s10546-007-9170-0, 2007.

---

## Author Comment (AC2) · 22 Jun 2017

**Authors' response to reviewer #2**

We thank reviewer # 2 for carefully reading our manuscript and the provision of many useful comments and detailed suggestions. We have considered all comments. They gave as useful hints where improvements of the paper were necessary to better understand our methodology and conclusions. Below all points raised by the reviewer are repeated; our comments are added in italics.

The changes (revised version vs. AMTD-paper) are highlighted as displayed by latexdiff ("diff.pdf", maybe renamed when uploaded as a supplement). For the sake of clarity only small changes are explicitly mentioned in our point by point replies, otherwise we refer to the corresponding parts of "diff.pdf" (in blue). Note, that some of our responses interact with comments of the other reviewers, so sometimes it is difficult to refer a change to one specific reviewer's comment.

**Point by point replies**

**General comment**

[...] Overall, the paper is well written and easy to follow, but however needs some more critical discussion on certain points.

In my point of view, using just one ceilometers/location might not be sufficient to answer the question given in the title. It is clear, that it is difficult to extent the study to other locations at that stage, but however, this aspect should be discussed more in detail. As highlighted by reviewer 1, I share the opinion that a day/night comparison might be interesting.

For meteorological conditions being a main driver of turbulent mixing it might be interesting to include some meteorological observations characterizing the measurement location and selected study period. With a observation height of about 5 m it might be interesting, which amount of the measured concentration is originated from the actual location and which amount is advected from neighboring areas or "removed" by vertical mixing. Here again a night/day difference would be interesting. How does this aspect influence on the analysis at one selected point?

> → *The first concern of reviewer #2 is the limited number of ceilometers and the missing discussion of day-night differences. This was also one*

*of the major criticisms of the first reviewer (please see our reply to reviewer #1 as well).*

*We agree with reviewer #2 that more ceilometers would have been beneficial for the study. It was however not possible to set up several ceilometers and/or to use mobile systems. Up to now ceilometers do not belong to the standard equipment of urban air quality networks, maybe this will change in future. So we had to rely on additional resources (in the framework of a campaign) with the inherent limitations (e.g. temporal availability).*

*Nevertheless we believe that BAERLIN2014 provided very valuable scientific results even if there was only one ceilometer available. We were able to demonstrate how differences in MLH-retrievals play a role for calculating correlations between MLH and air quality parameters. By addressing standard retrievals (the proprietary software of the ceilometer manufacturer) and air quality measurements from an official monitoring network we think that the conclusions are relevant. Based on our research, open questions could be identified one of which being the need for investigations of the variability of the mixing layer over a large municipality. So we hope that in future the wishes of the reviewer (and ours) to have more ceilometers and at least one full annual cycle of the MLH can be fulfilled, and that our paper will be a motivation for setting up the corresponding infrastructure (see also our replies to the detailed comments of reviewer #2 below).*

*As described also in the reply to reviewer #1 our conclusions on the large spatial variability of correlations between MLH and $PM_{10}$ are confirmed if we restrict our discussion to the stations nearby (less than 6.4 km from the ceilometer site). Over this small spatial domain the representativeness of a single MLH retrieval is very likely. Our discussion on the correlations between MLH and $NO_x$-concentrations also remain valid when focussing only on the vicinity of the ceilometer site.*

*As a consequence we have added a new paragraph (see pages 23–24 of diff.pdf) and more references (see page 9 of diff.pdf). Following the suggestions of reviewers #1 and #2 we have extended Sect. 5.1 by discussing day-night differences and the influence of the wind field (see comment of reviewer #3) (see pages 24–26 of diff.pdf). In addition a short comment on differences between working days and weekends has been added.*

*A detailed discussion of the influence of the local sources to measure-*

*ments in 5 m height is beyond the scope of this paper (see the clarification of our objectives in the introduction: page 3, line 9 ff of diff.pdf): such small scale investigations require much better temporal and spatial resolution of the measurements (and associated models). For example, station #42 used for the BAERLIN2014 project as reference is being classified as urban background station. This determines major pollution sources such as major streets to be not within the direct surrounding area (>100 m) and includes usually residential areas. Therefore minor sources like smokers, restaurants, barbecue, and household sources determine the moderate emissions in the vicinity. A moped or car passing the station for a short period of time is not detectable in an averaging period of one hour. Note, that the altitude of the ceilometer (5 m above ground) is not relevant for the determination of the MLH (see also comment on "p.8 line 1" below).*

**Specific comments**

- p.2 line 3-4: Is there evidence in your study? Otherwise put this sentence in the introduction or conclusion.

  → *From our point of view this message is important. That was the reason why it was included in the abstract. Note, that we have written "seems to be unrealistic ... a city like Berlin". It is not meant as a statement that is valid for all metropolitan areas worldwide at any time (for this we indeed do not have evidence); e.g., for cities surrounded by (high) mountain ridges or extreme pollution episodes the situation might be different. The sentence should be understood as a "warning" not to over-interpret correlations between MLH and concentrations of pollutants. To make this clearer we have modified the sentence in the following way: "seems to be unrealistic ... a city like Berlin (flat terrain)", and we have extended the introduction to better explain the scope of the paper (page 3 of diff.pdf).*

- p.2 line 19: measurements, data instead of techniques

  → *Improved*

- p.2, line 27: box models

$\rightarrow$ *Corrected*

- p.3, line 16: COBOLT: add one sentence highlighting novelty, functionality

  $\rightarrow$ *We have added a much more detailed description of COBOLT according to the suggestion of reviewer #1, see* pages 11-14 of diff.pdf.

- p.3, line 19: aim to instead of may

  $\rightarrow$ *We don't want to change this. The reason is that our study aims to show the influence of the retrieval on the derived MLH and the heterogeneity of the concentrations and thus may help the user to draw conclusions. We don't aim to show a link between air quality and MLH because there are more variables than just the MLH that control pollutant concentrations (see several comments of all reviewers and the statements in our manuscript). However, we have substituted "assess" by "interpret"* (page 4, top, of diff.pdf).

- p.4, line 1: specify "active remote sensing networks" (e.g:...)

  $\rightarrow$ *We have added* (e.g. the above mentioned ceilometer networks); page 4 line 19 of diff.pdf. *This refers to the introduction where we have added* (e.g. almost 100 instruments by the German Weather Service) *(see* page 3 of diff.pdf).

- p.5, line 15: chemical processes? What about Ozone? Where does it come from – downward mixing, secondary formation?

  $\rightarrow$ *In this section of the paper (introduction) we give an overview over previous publications that are relevant for our study. In the Schäfer et al. (2012) paper ozone was not considered, thus, it is not mentioned here. However, later in our paper we discuss this issue (Sect. 5.2): downward mixing, destruction of ozone by sometimes high $NO_x$ concentrations, production of ozone when $NO_x$ levels are low because of the notable amount of green spaces (parks, forests and leisure areas), or ozone formation by photochemistry* (page 27, lines 17 ff, of diff.pdf)

- p.5 line 28-31: This is not part of your analysis and could be moved to the conclusion.

$\rightarrow$ *Thanks for this remark: we agree and delete this part, as these ideas have already been included in the conclusions (so it was sort of a duplication).*

- p.6, line 9: specify "secondary material"

  $\rightarrow$ *We changed "secondary material" to "secondary aerosol compounds", see page 6, line 30 of diff.pdf*

- p.6, line 14: hourly measurement

  $\rightarrow$ *Corrected*

- p.8, line 1: how representative is the measurement location in 5 m height for near surface pm10 concentration? How does this impact the representativeness for the MLH measurements for this area?

  $\rightarrow$ *The ceilometer was installed 5 m above the ground. For the determination of the MLH a change of the altitude of the ceilometer in the range of a few meters is not relevant. Concentrations are measured at the BLUME stations approximately 3.5 m above ground. These values are expected to differ from measurements directly at the curbsite (see Bonn et al., (2016)). The latter might show much higher temporal fluctuations (e.g., passage of a car). Such microscale effects are not considered when correlations with the MLH are investigated. To resolve these problems certainly models at the building-resolving scale help. Moreover, during the transport from e.g. major traffic sites to the reference location strong vertical gradients will be smoothed (see also reply to "General Comment"). We have now briefly touched the topic of "scales" in the conclusions (page 31 of diff.pdf).*

- p.16 Figure 5: legend has to be added

  $\rightarrow$ *Done*

- p.17 line 2 ff: this chapter defines the scope of the study and in my opinion appears to late in the manuscript which results in a misbalance between introduction/methods and results. The first part until 5.1 is more an introduction to a new topic than a presentation of results. I might be helpful to include some of these aspects in the introduction

(without changing the whole manuscript). Line 2: Ozone and NOx also measured at BLUME?

    → *Thanks for this suggestion. Indeed this paragraph does not fit here very well. We have completely rephrased and re-arranged this paragraph. We removed text that was not relevant for our study. We moved the modified text to section 3 (new caption: "The BLUME network and the BAERLIN2014 campaign"). Now, all information related to the underlying data sets are combined in one section. We have also explicitly mentioned the distance of the BLUME-stations to the ceilometer as this is an essential point in view of the correlations discussed later (end of Sect. 3, page 9 of diff.pdf).*

    *Ozone and $NO_x$ are also measured by the BLUME network. This becomes clearer after moving text from Sect. 5 (introductory remarks) to Sect. 3, see above (from page 21 of the AMTD-version to page 7 of diff.pdf).*

    *Moreover, we introduced a new paragraph to the introduction to make the scope of our study more clear (see also reviewer #1; page 3 of diff.pdf).*

- p.18 line 8: it is unclear on which basis the median was calculated. 67 measurements each hour at every station?

    → *Yes, this is true for the concentration measurements at the BLUME-stations: the temporal resolution is one hour, and the whole measurement period of 67 days (i.e., when co-incident $PM_{10}$ and MLH measurements were available) is considered. With respect to the MLH we rely on all available 10-minutes retrievals (up to six, depending on the MLH-retrieval) of the corresponding hour, for all 67 days. So, up to 402 MLH-values are considered for the MLH-median. An new paragraph has been added to the end of Sect. 4.4 (pages 19–20 of diff.pdf).*

- p.18, line 16-20: can you proof your assumptions by adding meteorological observation here? Is there a secondary circulation generated by the Urban Heat Island? Please specify the term "meteorological interpretation".

    → *We are aware that we missed to clearly outline the scope of the*

*paper as necessary. We have corrected this now by adding a new paragraph to the introduction (page 3 lines 10 ff of diff.pdf).*

*In this context the term "meteorological interpretation" should be understood as the interpretation of processes that control the development of the mixing layer and surface concentration – and their interaction. A thorough discussion of the meteorological reasons and atmospheric chemistry responsible for the observed distribution of pollutants was however not the goal of the study. Nevertheless we have included several comments to point at reasons for poor or unexpected correlations.*

*Finally we want to emphasize that we present diurnal cycles of MLH and concentrations averaged over 67 days. The analysis of the interactions between meteorological fields (e.g. wind), atmospheric chemistry and emissions should rather be carried out with a high temporal resolution. This analysis would certainly benefit from a "complete" set of observations. Such a data set is however unrealistic. Thus, tentative answers may be found by numerical models. But models do not necessarily display proof of understanding or concepts but provide a further tool and support understanding. Anyway, such investigations are far beyond the scope of this paper (see a short comment in the conclusions page 31 of diff.pdf).*

- p.20, line 31f: see comment above

    → *See previous reply*

- p.21 general: here you mention briefly the problem of point measurements. This aspect could be further discussed. It is interesting if there is a mismatch between the timing of MLH and air quality observation. Does a low MLH mean a high concentration at the same time? What is the order of the processes? Where do meteorological conditions come into play?

    → *The answer to this question is closely related to the previous replies. It is not unlikely that a temporal delay between MLH and concentrations might occur, however, this delay is influenced by e.g. the wind field (upwind/downwind, low/high wind speed) or specific characteristics of the traffic (emissions). In case of secondary produced constituents it depends on the concentration*

*of the precursors and the solar irradiance. These influences are certainly time dependent, so it is hardly possible to detect them when long temporal averages are considered as in our study. We have briefly discussed the influence of the wind speed at the end of Sect. 5.1 by adding several paragraphs and Table 3 (pages 24–26 of diff.pdf).*

- Chapter 6: It is not the extended mixing layer itself which is the initial precursor of dilution of pollutants near the ground. Several processes interact which each other which might as well lead to an extension of the mixing layer height.

  → *We agree with the reviewer. We have mentioned the complex distribution of pollutants several times in the paper. Our goal was to compare this complexity with different schemes to determine the MLH from ceilometer data.*

---

## Author Comment (AC3) · 22 Jun 2017

**Authors' response to reviewer #3**

We thank reviewer # 3 for carefully reading our manuscript and the provision of many comments and suggestions. They gave as useful hints where improvements of the paper were necessary to better understand our methodology and conclusions. However, some of the raised questions have already been discussed in the submitted paper at different places, and some are clearly beyond the scope of the paper and/or cannot be resolved with the available data sets. Nevertheless we have considered all comments of reviewer #3. Our replies are given in italics.

The changes (revised version vs. AMTD-paper) are highlighted as displayed by latexdiff ("diff.pdf", maybe renamed when uploaded as a supplement). For the sake of clarity only small changes are explicitly mentioned in our point by point replies, otherwise we refer to the corresponding parts of "diff.pdf" (in blue). Note, that some of our responses interact with comments of the other reviewers, so sometimes it is difficult to refer a change to one specific reviewer's comment.

**Point by point replies**

**General comment**

This is an interesting manuscript as it discusses the relationship of the mixing layer height (MLH) and near surface pollutant concentrations. The authors perform correlations of the MLH and PM10, NOx, and O3, and found varying results. The authors believe that the effects of the heterogeneity of the emission sources, chemical processing and mixing during transport exceed the differences due to different MLH retrievals. With regard to the use of the different MLH retrieval methods (Vaisala proprietary software, COBOLT), which are solely based on aerosol backscatter signal, I was wondering, if radiosondes have been used for a conclusive validation during the BAERLIN campaign.

> → *Intercomparisons between aerosol-based MLH retrievals (lidars, ceilometers) and retrievals based on temperature-, wind- or water vapor profiles (e.g. from radio sondes) have been carried out in several studies; some papers have been cited in the manuscript. COBOLT has been developed using ceilometer measurements in Munich and compared with radio sondes data of Oberschleißheim (distance 8 km only). So we don't feel that it is necessary to demonstrate this again in this*

*paper. Moreover, it was not a goal of BAERLIN2014, and the closed radio sonde station is in Lindenberg, almost 60 km away from the ceilometer site!*

Also, I was wondering why other methods such as the Haar wavelet method or a cluster method have not been considered/discussed.

> → *The Haar wavelet method is one component of COBOLT when the Sobel operator is applied (see new citation of Comeron et al., 2013). This is mentioned in the revised version of the manuscript when we provide a much more detailed description of COBOLT (according to a suggestion of reviewer #1, pages 11–14 of diff.pdf). Moreover, we had already included three citations in the original manuscript (Cohn and Angevine, 2000, Brooks, 2003, Baars et al., 2008) that use this wavelet covariance transform. Caicedo et al. (2017) who applied the cluster method are cited as well (see response to reviewer #1).*

With regard to the relation of the MLH with air quality, it is well known that the local change of any given pollutant is not only controlled by the MLH, but by a combination of emission, chemical transformation, removal, advection, convection and turbulent mixing. Also, it is known that at the microscale level urban structures cause flow disturbations and thus deviations from the mean air quality of a larger, representative fetch in an urban area. An example is the well-known wind rotor system in street canyons. Thus the relationship of the MLH with surface concentration critically depends on the fetch area representative for a given measurement site. These well-known processes are not properly addressed in the paper.

> → *We agree with the reviewer that surface concentrations of pollutants do not only depend on the MLH and that our paper is not the first that points out these facts. Accordingly we have mentioned these processes e.g. in Sect. 5.1 (p 20, l27 ff. of the AMTD-version, p 20, l32 ff. including citations) and in the conclusions (p 25, l29 of the AMTD-version). As a consequence of the comments of reviewer #3 we have extended this discussion. Moreover, we have added a paragraph to the introduction where we describe the objectives of our study more clearly (see page 3 of diff.pdf). This was indeed not clear enough in the submitted manuscript: we want to focus on the ceilometer retrieval and the potential over-interpretation of correlations. These aspects have not yet been covered in the literature.*

*The reviewer's statement on the influence of "microscale level urban structures" certainly points out a very important aspect, which in sum would however have resulted in exploding project costs. Some aspects e.g. of chemical transformation and deposition can be reasonably well while not perfectly described by a chemical boxmodel. However, the vertical mixing aspect in such a model, determined by the MLH cannot be reproduced acceptably well without observations. The information provided by BAERLIN2014 supplies the effect of the MLH and therefore vertical exchange to the change of pollutants, i.e. the fraction of change that can be explained by meteorology.*

Due to the rather flat larger area of Berlin, it can be expected that transport processes may play a dominant role in the distribution of pollutants, both at the mesoscale and microscale level. I am surprised to see that the authors did not consider any of the findings associated with the BERLIOZ experiment in 1998 (mostly published in Journal of Atmospheric Chemistry 42, 2002, but also others), which focused on the upwind-downwind conditions found for the Berlin case, as well as the pollutant concentrations within the boundary layer and aloft in the same area and the impact of long-range transport.

→ *The BERLIOZ campaign (Berlin Ozone Experiment) was a huge campaign focussed on the impact of Berlin on the surroundings. It never investigated Berlin itself but a northwest-southeast transect through Berlin approximately 50 km on either side in Brandenburg, such as e.g. Pabsthum. In contrast the focus of BAERLIN2014 was the metropolitan area of Berlin and Potsdam and the influence of vegetation inside this area in detail. Thus, one could use references and results of BERLIOZ for broader discussions only.*

*Reviewer #3 seems to focus rather on large scale effects than on small scale mixing. Berlin is not affected by the surrounding countryside, somewhat more the opposite. This actually caused the BERLIOZ project nearly to fail, because the anticipated effects were hardly found (e.g. huge ozone plumes downwind, large $PM_{10}$ clouds etc.). As stated earlier the idea of the reviewer to conduct investigations for Berlin and Brandenburg including (very) detailed experiments and modelling approaches is far off realistic financial and personnel limits.*

*In conclusion, we don't feel it necessary to include any outcome from BERLIOZ as the scientific objectives of that experiment were quite different from our study. Neither the derivation of the MLH from*

*ceilometer measurements was part of BERLIOZ nor the distribution of pollutants inside the city.*

I would not expect an unambiguous relationship between the MLH and surface concentrations at any given location and under any given meteorological situation in the Berlin area. Rather, I would only expect a dominant role of the MLH on surface concentration, when advection is at a minimum, i.e. under stagnant wind conditions. In its current form this paper neglects the discussion of the MLH with regard to different wind regimes, both with regard to wind speed as wind direction. It should also be mentioned that not only pollutants can be transported, but also physical properties of the boundary layer including the MLH depending on the history of air masses. This extended in-depth analysis is a crucial requirement for a potential publication in AMT.

→ *We agree with the reviewer that advection plays a relevant role for the correlation between MLH and concentrations. This was briefly mentioned in the manuscript (see answer above). We have also elaborated this aspect in more detail in the revised version (pages 24–26 of diff.pdf) taking into account wind measurements of the German Weather Service at three sites in Berlin. We use these additional data set to select days when the wind was "predominantly stagnant". However, we want to emphasize that our mean diurnal cycles (MLH, concentrations) are averages over two months. So, the assessment of the contribution of a single process to the correlation between MLH and surface concentrations is hardly possible.*

**More specific, mostly minor issues**

- Page 2, L25-28: The paper by Czader et al. (2013) should be added as it is one of the earlier examples to use ceilometer derived MLHs for validation in conjunction with comprehensive air quality modeling.

  → *In Czader et al. (2013) we only find the reference to "remote sensing techniques" providing MLH at one site (Moody tower). Details were however found in Haman et al., 2012: here, CL31 measurements of almost two years have been evaluated for the diurnal cycle of the MLH in Houston, Texas. They use proprietary software of Vaisala. We have added both citations (pages 3 and 6).*

- Page 5, L18-19: I think both terms MLH and Hml mean the same. I suggest to use one term throughout the entire text.

  → *Our idea was to use MLH as "word" in the text, and $H_{ml}$, $H_{ml,v}$ etc. for mathematical expressions. We have checked this for consistency and changed it whenever necessary.*

- Page 5, L22: Please define what "width" would mean exactly: horizontal or vertical?

  → *Width is related to the MLH as derived from ceilometer measurements. As this could be misunderstood we changed the sentence to ...into intervals of 200 m. (see page 6 of diff.pdf).*

- Page 6, L9: Please define what is meant exactly by "secondary material"?

  → *We changed "material" to "secondary aerosol compounds", see page 6 of diff.pdf*

- Page 7, L1-3: "These data....in whole Germany". Is this statement important in understanding the contents of the paper? I suggest to remove it.

  → *We removed it as it is indeed not essential for the understanding. Anyway, for me it was an interesting information showing the extent of automatic air quality stations currently operational in Germany (see page 7 of diff.pdf).*

- Page 8, L4: What "information" is exactly meant?

  → *We have clarified this sentence: ...the option to combine in-situ measurements at the surface with data concerning the vertical direction (see page 9 of diff.pdf). The combination with aerosol optical depth would be another example. MLH is also useful to constrain model calculations as mentioned (see page 2 of diff.pdf).*

- Page 8, L6: Suggest to remove ", which is one hour different to UTC.", as UTC is not being used in the paper.

→ *This was included only as a explanation for readers who are more familiar with GMT. But we agree that it can be removed (see* page 9 of diff.pdf*).*

- Page 10, L16: Please explain what is actually meant by "cross-platform" here, and why it would be helpful?

  → *We wanted to emphasize that the code can be run on Windows and Linux platforms, so it is potentially useful for a large community. Moreover, Phyton is free of charge in contrast to e.g. MatLab. We have extended the whole section in accordance with the comments of reviewer #1; in this context we have also considered the comments of reviewer #3 (*pages 11–14 of diff.pdf*).*

- Page 15, L26: "Concentration measurements" of what?

  → *This could be "everything". In our study concentrations ($PM_{10}$, $NO_x$, $O_3$) are discussed but – if the corresponding data sets are available – the statement is also true for any other trace gas or e.g. $PM_{2.5}$. The sentence should only emphasize that problems may occur if data sets with low temporal resolution are considered during the rapid growth of the ML. To make this clearer we have substituted one word (*page 18, line 21 on diff.pdf*).*

- Page 15, L30: "The latter...(Pappalardo et al., 2014)". Please explain the schedule of EARLINET and explain whether the BAERLIN approach was important for the EARLINET approach or the other way round (which is more likely).

  → *The EARLINET schedule was defined in the year 2000. On the one hand it considers the diurnal cycle of the ML (measurements when the vertical extent is approximately constant for several hours) and on the other hand the performance of Raman lidars (they perform better during night). This was not influenced by our study, and our study is independent of the EARLINET schedule as we determine the full diurnal cycle. We only mentioned this because our (and similar) results confirmed that the selection made by EARLINET was reasonable (see COBOLT-retrieval shown in Fig. 5). For further illustration we have included Fig. 6. It shows the differences of the afternoon values of MLH when different MLH-retrievals are applied.*

*To make this clearer we have modified the corresponding sentence as follows: The latter has been the reason for including a measurement around two hours after local noon in the regular EARLINET schedule (page 18 of diff.pdf).*

- Page 15, L32-33: Please mention that these specific COBOLT results refer to the entire campaign period.

  → *We added ...for the whole period of 67 days (page 18 of diff.pdf).*

- Page 17, L11-12: What is exactly meant by "All measurements are performed under ambient conditions"? They way it is written it would mean that the air quality station was not air-conditioned.

  → *According to a comment of reviewer #2 we completely rephrased and re-arranged this paragraph. In this context we have also considered the comments of reviewer #3 and removed things that were not relevant in the context of our study (see page 7 of diff.pdf and answer to "Page 17, L15-16" below.*

- Page 17, L18: I think this "significant horizontal heterogeneity" refers to surface measurements here. Please clarify.

  → *Yes. Most of the measurements were made from bicycles. We have clarified this: Episodic mobile (bicycle) measurements from BAERLIN2014...(page 21 of diff.pdf)*

- Page 17, L15-16: What is exactly meant by "inorganic species": gasphase, particle bound or both?

  → *Inorganic species refer to gaseous compounds like CO, NO and $NO_2$. The whole paragraph has however been rephrased and reorganized according to a suggestion of reviewer #2. Main parts have been moved to Sect. 3 (from page 21 of diff.pdf to page 7 of diff.pdf), and unnecessary information was deleted.*

- Page 17, L17: The reference "von Schneidemesser et al., 2017" is still in preparation and therefore not citable.

  → *We removed this citation and the text (lines before this citation) that was related to this paper which is currently still under preparation.*

- Page 17, L17-18: "Here we do not discuss these topics...". In this case please remove the preceding L13-17 as they are not within the scope of this paper.

    → *See our above response to the comment on "Page 17, L11-12"; the sentence was removed.*

- Page 18, L26-28: What is the justification for using these different correlations? The statistically most reliable quantity would be the median anyway, as it minimizes the impact of outliers. This is in particular true for such a quantity as PM10, which is mostly primarily emitted.

    → *We agree with the reviewer: that was the reason why we use the median in Figs. 7, 9 and 10 in the AMTD-manuscript. The same argumentation as the comment of the reviewer was given in the original manuscript (page 20, lines 6 ff). The different combinations of averages and medians as defined on page 18, lines 26 ff were only introduced to demonstrate the consequences on the correlations in the subsequent discussion. See also the new Fig. 7 (page 20 of diff.pdf).*

- Page 20, L23: "...with a lot of vegetation, a high density of buildings...". This sounds like a contradiction: where there is high density of buildings how can there be lot of vegetation at the same time?

    → *This description is made from the perspective of a German citizen. A "high density of buildings" does not mean that there is no space left for trees, bushes etc., often arranged as small "parks" of some tens of meters in length and width, or buildings organized as squares with trees inside a yard, to increase the quality of living. For example, southeast of the ceilometer is an area of approximately $100 \times 70$ m with "a lot of vegetation" whereas buildings dominate elsewhere. To avoid misunderstands we replace "high density of buildings" by "in a typical residential neighborhood in the inner part of a big German city; see page 24 of diff.pdf". A similar expression has also been used in Sect. 3.*

- Page 20, L17-18: The authors mention aerosol formation. Would PM10 data provide any indication for aerosol formation? If so, please explain.

$\rightarrow$ *When we summarized our conclusions from Fig. 9, we mentioned different aspects that are responsible that no unique correlation coefficient (MLH vs. $PM_{10}$) has been found for entire Berlin. In this context only the absolute value of $PM_{10}$ is relevant.*

- Page 20, L17: The authors mention that relative humidity may have an impact on PM10. Would PM10 concentration decrease or increase with relative humidity?

  $\rightarrow$ *The whole paragraph was significantly extended (see also replies to Reviewers #1 and #2) by including more investigations on correlations under special meteorological conditions (see pages 24–26 of diff.pdf). In this context the statement on the relative humidity became unnecessary (one can assume a small increase due to uptake of $HNO_3$).*

- Page 21, L1-2: What classes in addition would the authors recommend?

  $\rightarrow$ *We do not necessarily need more classes but the attribution might be reviewed. However we don't have any influence on this classification and the criteria for this classification. The same is true for the selection of the locations of the air quality stations. It is not unlikely that political reasons have a certain influence as well. We have added a short remark at the end of Sect. 5.1 (page 26 of diff.pdf).*

- Page 21, L2-4: This statement is obvious and has been considered in many urban air quality networks over many decades.

  $\rightarrow$ *This conclusion is indeed not unexpected. Nevertheless many publications do not clearly describe the conditions under which their correlation has been calculated or use only one site in a metropolitan area and leave it open how representative their conclusions are. So there remains room for misunderstandings, and we feel that it is justified/necessary to emphasize this statement (again). Accordingly we have expressed this objective in the updated introduction (see page 3 of diff.pdf).*

- Page 21, L31 - Page 23, L5: It is well-known that O3 can be mixed from the residual layer into the convective layer, also for the case of

Berlin (e.g. see BERLIOZ special issue in the Journal of Atmospheric Chemistry 42, 2002). The excellent correlation of the MLH with ozone in urban areas may not be surprising at all, as both processes are ultimately driven by incoming solar radiation provided there are sufficient precursors for O3 formation available. In other words the relation between the MLH and O3 is apparent, but not causally determined. This should be mentioned.

→ *We agree with the reviewer: we have used the same argumentation in that paragraph of the AMTD-version of the manuscript including the citation of a paper by Fallmann et al.; so it has already been mentioned. To make this clearer we slightly rephrased this paragraph (see page 27 bottom of diff.pdf).*

• Page 24, L7: "Whether ...studied". I suggest to remove this sentence. It is obvious that the potential impact of the MLH on ambient concentrations decreases with decreasing distance to the corresponding emission source.

→ *We have removed this sentence.*

• Page 24, L13: As I remember Xu et al. (2011) do not report MLH observations and thus no correlation with primary or secondary pollutants.

→ *Xu et al. (2011) discussed the influence of the MLH on surface concentrations of several trace gases in a general way. However, they did not use own measurements of the MLH. As a consequence we agree with the reviewer that this citation is not really relevant and dropped it.*

• Page 25, L10-12: I guess it is well-known that there is not one only parameter which controls surface concentrations.

→ *Again we agree with the reviewer, but our motivation was to investigate the role of the MLH-retrieval for correlation studies in view of its uncertainty and the inhomogeneity of urban air quality. This message was obviously not as clear as it should have been (see corresponding comments of all reviewers). As a consequence we have added a clear statement on our objectives to the introduction (see page 3 of diff.pdf).*

- Page 25, L27-28: In their paper the authors have tried to argue that MLH is not the only parameter which controls surface pollutant concentrations. Why then would it be of interest to perform a winter study in Berlin and why is it of importance that PM10 concentrations are 50% higher in winter compared to summer in Berlin. If there is no consistent correlation of MLH with PM10 in summertime why should it be different in wintertime?

  → *Reviewers #1 and #2 regret that only data from two months were available. This cannot be changed for obvious reasons. However, we believe (together with the reviewers) that a longer observation time would provide additional insight: in winter the MLH is expected to be shallower, the concentration of $PM_{10}$ is larger, and the meteorological conditions (including atmospheric chemistry) are different. We do not expect that in winter the MLH is the only parameter that controls the concentration of pollutants, but it is not clear if the variability of R is more or less pronounced. It is scientific tradition to investigate any problem under different conditions if possible (see our comments on available resources). To point this out we have added an additional explanation: We do not expect that in winter the MLH is the only controlling parameter, but it is not clear if the correlation (and its variability) is more or less pronounced (see page 31, lines 17 ff of diff.pdf).*

- Page 26, L4-6: The authors state that MLH data is beneficial for box-model calculations and validation of chemistry transport models. While I would agree on the authors statement in this sentence I do not completely understand what the authors justification would be for this, since according to their paper the authors largely argue that there is no consistent correlation of the MLH with air pollutants. This should be clarified.

  → *In any case there are multiple counteracting processes merging in our findings as has been mentioned in the paper. As a consequence interpretation is much more complex than simply getting $R \approx \pm 1$ for all times, but this does not reduce the usefulness of a reliable determination of the MLH. This was our statement in the last paragraph of our conclusions. We have clarified this by extending the conclusions on validation and combination of models and measurements (see page 31, bottom, of diff.pdf). It would,*

*e.g., be nice to tackle the question of the homogeneity of the MLH over a city like Berlin by models and compare the results with distributed ceilometer measurements (see our corresponding replies to similar questions of all reviewers), maybe possible in future. Moreover, model calculations can help to understand the interaction and the relevance of different meteorological and chemical processes; in this context it could be useful to have independent measurements to validate at least parts of the model output (again a question of resources to set up a adequate field campaign).*

Additional references (for more see also reply to reviewer #1):

- Czader, B. H., Li, X., and Rappenglueck, B.: CMAQ modeling and analysis of radicals, radical precursors and chemical transformations, J. Geophys. Res., 118, 11,376–11,387, doi: 10.1002/jgrd.50807, 2013.

- Haman, C. L., Lefer, B., and Morris, G. A.: Seasonal Variability in the Diurnal Evolution of the Boundary Layer in a Near-Coastal Urban Environment, J. Atmos. Oceanic Technol., 29, 697710, 2012.

---

## Author Comment (AC5) · 22 Jun 2017

This file (AC4) does not include all modifications by reviewers #1, #2 and #3, but our modifications triggered by the comments of reviewers #1, #2 and #3. Sorry for the confusion.

---

## Referee Report (RR1)

Report AMT-2017-53 –Mixing layer height as an indicator for urban air quality?

The authors have extensively answered to the comments of the reviewers and have significantly improved the manuscript. Of course, the inherent limitations as the short measurement period and the presence of only one ceilometer could not be changed. However, there is enough interesting content that the paper can now be published; there are only 2 suggestions for minor corrections (see below). The strength of the paper is the comparison of the BL-VIEW and COBOLT algorithms to determine MLH from the ceilometer and the profound analysis of the air pollution data with respect to MLH.

Suggestions for corrections:

p. 1, line 7: … conducted during June and August 2014: better replace by … conducted from June to August 2014.

p. 2, line 4: e.g. as input to dispersion models and for the validation of ….

---

## Author Response (AR2)

**Authors' response to reviews of the revised version**

We thank the reviewers for carefully reading the revised manuscript and their positive comments. The two suggestions of reviewer #1 to improve the abstract has been considered. Moreover, we have included another reference to the manuscript:

Kaser, L., E. G. Patton, G. G. Pfister, A. J. Weinheimer, D. D. Montzka, F. Flocke, A. M. Thompson, R. M. Stauffer, and H. S. Halliday (2017), The effect of entrainment through atmospheric boundary layer growth on observed and modeled surface ozone in the Colorado Front Range, J. Geophys. Res. Atmos., 122, 60756093, doi:10.1002/ 2016JD026245.